# SURREAL-GAN: SEMI-SUPERVISED REPRESENTATION LEARNING VIA GAN FOR UNCOVERING HETEROGENEOUS DISEASE-RELATED IMAGING PATTERNS

**Zhijian Yang[1,2], Junhao Wen[1] and Christos Davatzikos[1]**
[1]Center for Biomedical Image Computing and Analytics, University of Pennsylvania
[2]Graduate Group in Applied Mathematics and Computational Science, University of Pennsylvania
`{zhijian.yang,junhao.wen,christos.davatzikos}@pennmedicine.upenn.edu`

## ABSTRACT

A plethora of machine learning methods have been applied to imaging data, enabling the construction of clinically relevant imaging signatures of neurological and neuropsychiatric diseases. Oftentimes, such methods do not explicitly model the heterogeneity of disease effects, or approach it via nonlinear models that are not interpretable. Moreover, unsupervised methods may parse heterogeneity that is driven by nuisance confounding factors that affect global brain structure or function, rather than heterogeneity relevant to a pathology of interest. On the other hand, semi-supervised clustering methods seek to derive a pathology-related dichotomous subtypes, ignoring the truth that disease heterogeneity spatially and temporally extends along a continuum. To address the aforementioned limitations, herein, we propose a novel method, termed Surreal-GAN (**S**emi-**SU**pe**R**vised **R**epr**E**sent**A**tion **L**earning via GAN). Using cross-sectional imaging data, Surreal-GAN dissects underlying disease-related heterogeneity under the principle of semi-supervised clustering [cluster mappings from the normal control (CN) to patient (PT) domain], proposes a continuously dimensional representation, and infers the disease severity of patients at individual level along each dimension. The model first learns a transformation function from the CN domain to the PT domain with latent variables controlling transformation directions. An inverse mapping function together with regularization on function continuity, pattern orthogonality and monotonicity was also imposed to make sure that the transformation function captures necessarily meaningful imaging patterns with clinical significance. We first validated the model through semi-synthetic experiments, and then demonstrated its potential in capturing biologically plausible imaging patterns in Alzheimer's disease (AD).

## 1 INTRODUCTION

Neuroimaging, conventional machine learning, and deep learning have offered unprecedented opportunities to understand the underlying mechanism of brain disorders over the past decades (Davatzikos (2018)), and pave the road towards individualized precision medicine (Bzdok & Meyer-Lindenberg (2018)). A large body of case-control studies leverage mass univariate group comparisons to derive structural or functional neuroimaging signatures (Habeck et al. (2008), Hampel et al. (2008), Ewers et al. (2011)). However, these studies suffer from under-powered statistical inferences since the heterogeneous nature of neurological diseases usually violate the assumption that each group population are pathologically homogeneous.

A developing body of methodologies aim to disentangle this heterogeneity with various machine learning methods. Zhang et al. (2016) used the Bayesian Latent Dirichlet Allocation (LDA) model to identify latent atrophy patterns from voxel-wise gray matter (GM) density maps. However, due to the nature of the LDA model, the density maps need to be first discretized, and the model hypothesized to exclusively capture brain atrophy while ignoring the potential regional enlargement. Young et al. (2018) proposed another method, Sustain, to uncover temporal and phenotypic heterogeneity by inferring both subtypes and stages. However, the model only handles around 10-20 large

brain regions without more detailed information. Notably, both methods applied directly in the PT domain, confront main limitations in avoiding potential disease-unrelated brain variations. In contrast, semi-supervised clustering methods (Varol et al. (2016), Dong et al. (2015), Wen et al. (2021), Yang et al. (2021)) were proposed to cluster patients via the patterns or transformations between the reference group (CN) and the target patient group (PT). A recent proposed deep learning-based semi-supervised clustering method, termed Smile-GAN (Yang et al. (2021)), achieved better clustering performance by learning multiple mappings from CN to PT with an inverse clustering function for both regularization and clustering. Albeit these methods demonstrated potential in clinical applications, they have a major limitation. In particular, these methods model disease heterogeneity as a dichotomous process and derive a discrete output, i.e., subtype, ignoring the fact that brain disorders progress along a continuum. Moreover, variations among subtypes were contributed by both spatial and temporal differences in disease patterns, thus confounding further clinical analysis.(Fig. 1a)

To address the aforementioned limitations, we propose a novel method, Surreal-GAN (Semi-Supervised Representation Learning via GAN), for deriving heterogeneity-refined imaging signatures. Surreal-GAN is a significant extension of Smile-GAN under the same principle of semi-supervised learning by transforming data from CN domain $\mathbb{X}$ to PT domain $\mathbb{Y}$. Herein, the key extension is to represent complex disease-related heterogeneity with low dimensional representations with each dimension indicating the severity of one relatively homogeneous imaging patterns (Fig. 1b). We refer to these low dimensional representations as R-indices ($\mathbf{r}_i$, where $i$ represent the $i_{th}$ dimension) of disease pattern. The first key point of the method is modelling disease as a continuous process and learning infinite transformation directions from CN to PT, with each direction capturing a specific combination of patterns and severity. The idea is realized by learning one transformation function which takes both normal data and a continuous latent variable as inputs and output synthesized-PT data whose distribution is indistinguishable from that of real PT data. The second key point is controlling the monotonicity (from light to severe with increasing index value) of disease patterns through one monotonicity loss and double sampling procedure. The third key point is boosting the separation of captured disease patterns through one orthogonality regularization. The fourth key point is introducing an inverse mapping function consisting of a decomposition part and a reconstruction part which further ensure that patterns synthesized through transformation are exactly captured for inferring representations of disease patterns, the R-indices. Besides the above mentioned regularization, function continuity, transformation sparsity and inverse consistency further guide the model to capture meaningful imaging patterns.

To support our claims, we first validated Surreal-GAN on semi-synthetic data with known ground truth of disease patterns and severity. We compared performance of Surreal-GAN to NMF (Lee & Seung (1999)), opNMF (Sotiras et al. (2014)), Discriminant-NMF (Lee et al. (2012)), Factor Analysis and LDA (Blei et al. (2001)) models, and subsequently studied the robustness of model under various conditions. Finally, we applied the model to one real dataset for Alzheimer's disease (AD), and demonstrated significant correlations between the proposed imaging signatures and clinical variables.

## 2 METHODS

The schematic diagram of Surreal-GAN is shown in Fig. 1d. The model is applied on regional volume data (volumes of different graymatter (GM) and whitematter (WM) regions) derived from structural MRI. The essential idea is to learn one transformation function $f : \mathbb{X} * \mathbb{Z} \to \mathbb{Y}$ which transforms CN data $\mathbf{x}$ to different synthesized PT data $\mathbf{y}' = f(\mathbf{x}, \mathbf{z})$, with latent variable $\mathbf{z}$ specifying distinct combinations of patterns and severity. Here, the Latent (LAT) domain, $\mathbb{Z} = \{\mathbf{z} : 0 \leq \mathbf{z}_i \leq 1 \, \forall 1 \leq i \leq M\}$, is a class of vectors with dimension $M$ (predefined number of patterns). We denote four different data distributions as $\mathbf{x} \sim p_{cn}(\mathbf{x})$, $\mathbf{y} \sim p_{pt}(\mathbf{y})$, $\mathbf{y}' \sim p_{syn}(\mathbf{y}')$ and $\mathbf{z} \sim p_{lat}(\mathbf{z})$, respectively, where $\mathbf{z} \sim p_{lat}(\mathbf{z})$ is sampled from a multivariate uniform distribution $U[0, 1]^M$, rather than a categorical distribution as in Smile-GAN. A continuous z variable lay the foundation for learning continuous representations. In addition to the transformation function, an adversarial discriminator $D$ is introduced to distinguish between real PT data $\mathbf{y}$ and synthesized PT data $\mathbf{y}'$, thereby ensuring that the synthesized PT data are indistinguishable from real PT data (i.e. minimizing the distance between real PT data distribution and synthesized PT data distribution).

A continuous latent variable does not necessarily lead to desired properties of CN to PT transformation (e.g. separation/monotonicity of captured patterns). That is, there are a number of functions potentially achieving equality in distributions, thus making it hard to guarantee that the transfor-

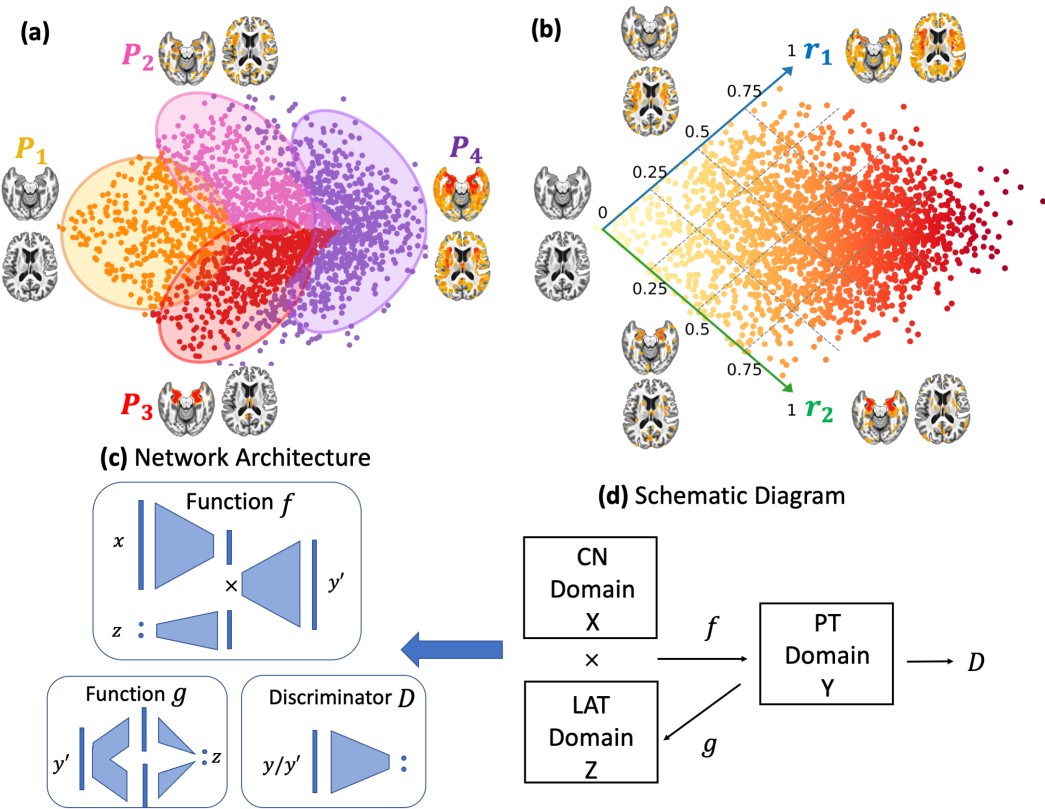

Figure 1: Model Ideas and Schematic Diagram. (a) Results from Semi-supervised clustering for AD disease: model clusters PT into discrete subtypes based on both spatial and temporal variations: P1 and P4 show mild and severe stages and are mixture of two patterns; (b) Improvements from Surreal-GAN: two patterns are captured by two indices with values indicating severity; (c) General architecture of networks; (d) Schematic diagram of Surreal-GAN.

mation function learned by the model is closely related to the underlying pathology progression process. Moreover, during the training procedure, the transformation function backboned by the neural network tends to ignore the latent variable $\mathbf{z}$. Even when the Latent variable $\mathbf{z}$ is emphasized through regularization, different components of $\mathbf{z}$ variable do not naturally lead to separate disease patterns and are not positively correlated with pattern severity. Therefore, with the assumption that there is one true underlying function for real PT data $\mathbf{y} = h(\mathbf{x}, \mathbf{z})$, Surreal-GAN aims to boost the transformation function $f$ to approximate the true underlying function $h$, while having the latent variable, $\mathbf{z}$, contributing to recognizable imaging patterns rather than being random "activators" of the GAN. For this purpose, we constrain the function class of $f$ via different types of regularization as follows: 1) encourage sparse transformations; 2) enforce Lipschitz continuity of functions; 3) introduce one inverse function $g : \mathbb{Y} \to \mathbb{Z}$ serving to decompose and reconstruct; 4) boost orthogonality among synthesized/captured patterns; 5) enforce monotonicity of synthesized patterns and their positive correlations with components of $\mathbf{z}$. The training procedure of Surreal-GAN is thus a adversarial learning process constrained by these regularization.

## 2.1 ADVERSARIAL LOSS

The adversarial loss (Goodfellow et al. (2014)) is used for iterative training of discriminator $D$ and the transformation function $f$, which can be written as:

$$L_{\text{GAN}}(D, f) = \mathbb{E}_{y \sim p_{pt}(\mathbf{y})}[\log(D(\mathbf{y}))] + \mathbb{E}_{\mathbf{x} \sim p_{cn}(\mathbf{x}), \mathbf{z} \sim P_{lat}(\mathbf{z})}[\log(1 - D(f(\mathbf{x}, \mathbf{z})))] \quad (1)$$

$$= \mathbb{E}_{\mathbf{y} \sim p_{pt}(\mathbf{y})}[\log(D(\mathbf{y}))] + \mathbb{E}_{\mathbf{y}' \sim p_{syn}(\mathbf{y}')}[\log(1 - D(\mathbf{y}'))] \quad (2)$$

The transformation function $f$ attempts to transform CN data to synthesized PT data so that they follow similar distributions as real PT data. The discriminator, providing a probability that $\mathbf{y}$ comes from the real data rather than transformation function, tries to identify the synthesized PT data and distinguish it from real PT data. Therefore, the discriminator $D$ attempts to maximize the adversarial loss function while the transformation function $f$ attempts to minimize against it. The training process can be denoted as:

$$\min_f \max_D L_{\text{GAN}}(D, f) = \mathbb{E}_{\mathbf{y} \sim p_{pt}(\mathbf{y})}[\log(D(\mathbf{y}))] + \mathbb{E}_{\mathbf{y}' \sim p_{syn}(\mathbf{y}')}[\log(1 - D(\mathbf{y}'))] \tag{3}$$

## 2.2 REGULARIZATION

### 2.2.1 SPARSE TRANSFORMATION

We assumed that disease process will not change brain anatomy dramatically and primarily only affect certain regions throughout most of disease stages, which means that the true underlying transformation only moderately changes some regions while keeping the rest mildly affected with most of $\mathbf{z}$ variable values. Therefore, to control sparsity and distance of transformations, we defined one change loss to be the $l_1$ distance between the synthesized PT data and the original CN data:

$$L_{\text{change}}(f) = \mathbb{E}_{\mathbf{x} \sim p_{cn}(\mathbf{x}), \mathbf{z} \sim p_{lat}(\mathbf{z})}[||f(\mathbf{x}, \mathbf{z}) - \mathbf{x}||_1] \tag{4}$$

### 2.2.2 LIPSCHITIZ CONTINUITY AND INVERSE MAPPING

First, by enforcing the transformation function $f$ to be $K_1$-Lipschitz continuous, we can derive that, for fixed latent variable $\mathbf{z} = \boldsymbol{a}$ and $\forall \boldsymbol{x}_1, \boldsymbol{x}_2 \in \mathbb{X}$, $||f(\boldsymbol{x}_1, \boldsymbol{a}) - f(\boldsymbol{x}_2, \boldsymbol{a})||_2 \leq K_1 ||\boldsymbol{x}_1 - \boldsymbol{x}_2||_2$. Thus, by controlling the constant $K_1$, we constrained the transformation function to preserve the original distances among CN data instead of scattering them into the PT domain if a same combination of disease patterns is imposed to them.

Second, to prevent latent variable $\mathbf{z}$ from being ignored, we introduced an inverse mapping function from PT domain to Latent domain, $g : \mathbb{Y} \to \mathbb{Z}$. By Lemma1 (section A.1), with function $g$ being $K_2$-Lipschitz continuous and $d(\cdot, \cdot)$ being any distance metric satisfying triangular inequality, we can derive that, $\forall \boldsymbol{z}_1, \boldsymbol{z}_2 \sim p_{lat}(\mathbf{z})$, $\boldsymbol{z}_1 \neq \boldsymbol{z}_2$, and $\bar{\boldsymbol{x}} \sim p_{cn}(\mathbf{x})$, $d(f(\bar{\boldsymbol{x}}, \boldsymbol{z}_1), f(\bar{\boldsymbol{x}}, \boldsymbol{z}_2))$ is lower bounded by $(\frac{d(\boldsymbol{z}_1, \boldsymbol{z}_2)}{K_2} - \frac{1}{K_2}(d(g(f(\bar{\boldsymbol{x}}, \boldsymbol{z}_1)), \boldsymbol{z}_1) + d(g(f(\bar{\boldsymbol{x}}, \boldsymbol{z}_2)), \boldsymbol{z}_2)))$. Therefore, by minimizing the distance between sampled latent variable $\mathbf{z}$ and reconstructed latent variable $g(f(\mathbf{x}, \mathbf{z}))$, we can control differences between synthesized patterns to be positively related to distances between latent variables, and thus to be non-trivial (i.e., same CN data is transformed to significantly different PT data with different $\mathbf{z}$ variables). Therefore, we define a reconstruction loss as the $l_2$ distance between $g(f(\mathbf{x}, \mathbf{z}))$ and $\mathbf{z}$:

$$L_{\text{recons}}(f, g) = \mathbb{E}_{\mathbf{x} \sim p_{cn}(\mathbf{x}), \mathbf{z} \sim p_{lat}(\mathbf{z})}[||g(f(\mathbf{x}, \mathbf{z})) - \mathbf{z}||_2] \tag{5}$$

### 2.2.3 PATTERN DECOMPOSITION

Nevertheless, a simple function $g$ directly reconstructing sampled $\mathbf{z}$ from synthesized PT $\mathbf{y}'$ do not reconstruct each component $\mathbf{z}_i$ completely based on changes led by $\mathbf{z}_i$ in transformation. We are more interested in pattern representations of real PT data than generating synthesized PT, and inverse function $g$ will be used for inferring R-indices as introduced in section 2.4. Thus, it is very important for the inverse function to accurately capture patterns synthesized by intensively regularized transformation function $f$. For this purpose, we further separated the function $g : \mathbb{Y} \to \mathbb{Z}$ into $g_1 : \mathbb{Y} \to \mathbb{R}^{M*S}$ (where $M$ is number of patterns and $S$ is number of input ROIs) and $g_2 : \mathbb{R}^S \to \mathbb{R}$. Decomposer $g_1$ serves to reconstruct changes synthesized by each component $z_i$ in transformation process: $\mathbf{q}_i = f(\mathbf{x}, \mathbf{a}^i) - \mathbf{x}$, where $\mathbf{a}^i$ is a vector with $i_{th}$ component $\mathbf{a}_i^i = \mathbf{z}_i$ and $\mathbf{a}_j^i = 0, \forall i \neq j$. Let $\hat{\mathbf{q}}_{f(x,z)} = [\mathbf{q}_1^T, \cdots, \mathbf{q}_M^T]^T$ be concatenation of all synthesized changes $\mathbf{q}_i$, we defined the decomposition loss as:

$$L_{\text{decom}}(f, g_1) = \mathbb{E}_{\mathbf{x} \sim p_{cn}(\mathbf{x}), \mathbf{z} \sim p_{lat}(\mathbf{z})}[||g_1(f(\mathbf{x}, \mathbf{z})) - \hat{\mathbf{q}}_{f(\mathbf{x}, \mathbf{z})}||_2] \tag{6}$$

$g_2$ serves to further reconstruct each component of sampled $\mathbf{z}$ variable from $g_1(f(\mathbf{x}, \mathbf{z}))$ independently, so that the reconstruction loss can be rewritten as:

$$L_{\text{recons}}(f, g) = L_{\text{recons}}(f, g_1, g_2) = \mathbb{E}_{\mathbf{x} \sim p_{cn}(\mathbf{x}), \mathbf{z} \sim p_{lat}(\mathbf{z})}[||\hat{\mathbf{l}}_{g_2(g_1(f(\mathbf{x}, \mathbf{z})))} - \mathbf{z}||_2] \tag{7}$$

where $\hat{\mathbf{l}}_{g_2(g_1(f(\mathbf{x},\mathbf{z}))} = g(f(\mathbf{x},\mathbf{z})) = [g_2(g_1(f(\mathbf{x},\mathbf{z}))_{0:S}), \cdots, g_2(g_1(f(\mathbf{x},\mathbf{z}))_{S*(M-1):S*M})]^T$.
Function $g$ here can be also considered as the approximation of expectation of posterior distribution. In this sense, minimization of $l_2$ distances can also be interpreted as maximization of mutual information between latent variable $\mathbf{z}$ and synthesized PT data $\mathbf{y}' = f(\mathbf{x}, \mathbf{z})$ as explained in Remark1 in section A.2 (Chen et al. (2016)). From this perspective, the transformation function is forced to best utilize information from latent variable $\mathbf{z}$, while also keeping mutual information between synthesized PT data $\mathbf{y}'$ and original CN data $\mathbf{x}$ by controlling transformation distances.

### 2.2.4 Orthogonality of Patterns

Inclusion of latent variable $\mathbf{z}$ in the transformation function does not guarantee that different components of the latent variable, $\mathbf{z}_i$, contributes to relatively different patterns in synthesized PT data. Instead, during training procedure, different components, $\mathbf{z}_i$, are more likely to synthesize patterns in same regions and lead to accumulated severity. Therefore, we define another orthogonal loss to encourage separation among patterns synthesized by different components. With changes led by each component, $\mathbf{q}_i$ defined in section 2.2.3, we constructed a matrix $\mathbf{A}_{f(\mathbf{x},\mathbf{z})}$ with the $i_{th}$ column $\mathbf{A}_{f(\mathbf{x},\mathbf{z})_{:,i}} = |\mathbf{q}_i|/||\mathbf{q}_i||_2$. To encourage separation of changes led by different components, we boosted the matrix $\mathbf{A}_{f(\mathbf{x},\mathbf{z})}$ to be relatively orthogonal by minimizing the following orthogonality loss:

$$L_{\text{ortho}}(f) = \mathbb{E}_{\mathbf{x}\sim p_{cn}(\mathbf{x}),\mathbf{z}\sim p_{lat}(\mathbf{z})}[||\mathbf{A}_{f(\mathbf{x},\mathbf{z})}^T \mathbf{A}_{f(\mathbf{x},\mathbf{z})} - \boldsymbol{I}||_F] \tag{8}$$

### 2.2.5 Monotonicity and Positive Correlation

Besides separation of synthesized patterns, a positive correlation between values of each component $\mathbf{z}_i$ (from 0 to 1) and severity of synthesized patterns can not be acquired spontaneously either. Regarding severity of the disease pattern, we assumed that, as a pattern is becoming more severe, absolute values of changes in regional volumes can only increase monotonically or remain constant. In another word, there is no oscillation in regional volumes as disease pattern is becoming more severe, while more regions can be included gradually with absolute changes switched from 0 to a positive value. To satisfy this requirement, we sampled another latent variable $\mathbf{z}' \sim p_{sev}(\mathbf{z}'|\mathbf{z})$, conditioned on previous sampled $\mathbf{z}$ variable, such that $\mathbf{z}'_i \geq \mathbf{z}_i, \forall 1 \leq i \leq M$. With these double sampled latent variables, we defined the monotonicity loss to be:

$$L_{\text{mono}}(f) = \mathbb{E}_{\mathbf{x}\sim p_{cn}(\mathbf{x}),\mathbf{z}\sim p_{lat}(\mathbf{z}),\mathbf{z}'\sim p_{sev}(\mathbf{z}'|\mathbf{z})}[||\max(|f(\mathbf{x},\mathbf{z})-\mathbf{x}| - |f(\mathbf{x},\mathbf{z}')-\mathbf{x}|, \mathbf{0})||_2] \tag{9}$$

Minimization of this monotonicity loss penalize the case that volume changes induced by $\mathbf{z}$ is larger than that induced by $\mathbf{z}'$, while the other direction is permitted. To further ensure a positive correlation (small $\mathbf{z}$ lead to mild patterns; large $\mathbf{z}$ lead to severe patterns), we introduced a cn loss, which serves to more strictly constrain the distance between synthesized PT data $\mathbf{y}'$ and CN data $\mathbf{x}$, when latent variable $\mathbf{z}$ is close to $\mathbf{0}$ vector. By letting $p_{cn}(\mathbf{z}) = U(0, 0.05)^M$ to be a multivariate uniform distribution, the cn loss is defined as:

$$L_{\text{cn}}(f) = \mathbb{E}_{\mathbf{x}\sim p_{cn}(\mathbf{x}),\mathbf{z}^{cn}\sim p_{cn}(\mathbf{z})}[||f(\mathbf{x},\mathbf{z}^{cn}) - \mathbf{x}||_1] \tag{10}$$

### 2.3 Full Objective

With the adversarial loss and all other loss functions defined for regularization, we can write the full objective as:

$$L(D, f, g_1, g_2) = L_{\text{GAN}}(D, f) + \gamma L_{\text{change}}(f) + \kappa L_{\text{decom}}(f, g_1) + \zeta L_{\text{recon}}(f, g_1, g_2)$$
$$+ \lambda L_{\text{ortho}}(f) + \mu L_{\text{mono}}(f) + \eta L_{\text{cn}}(f) \tag{11}$$

with $\gamma$, $\kappa$, $\zeta$, $\lambda$, $\mu$ and $\eta$ be parameters controlling the relative importance of different loss functions. Through training process, we want to derive parametrized functions $f$, $g_1$ and $g_2$ satisfying:

$$f, g_1, g_2 = \arg\min_{f,g}\max_{D} L(D, f, g_1, g_2) \tag{12}$$

### 2.4 Representation-Indices Inference

With the assumption that $p_{syn}(f(\mathbf{x}, \mathbf{z})) \approx p_{pt}(\mathbf{y})$ and $f$ satisfying all constraints, we consider learned transformation function $f$ to be a good approximation of the true underlying function $h$, such

that $f(\mathbf{x}, \mathbf{z}) \approx h(\mathbf{x}, \sigma(\mathbf{z}))$, where $\sigma \in \Omega$ and $\Omega$ is a class of permutation functions which change orders of elements in vector $\mathbf{z}$. Since the order of indices in derived representation is not important and we can always reorder them to find the best matching, we simply rewrite it as $f(\mathbf{x}, \mathbf{z}) \approx h(\mathbf{x}, \mathbf{z})$ without loss of generality. For any real PT data, $\bar{\mathbf{y}} = h(\bar{\mathbf{x}}, \mathbf{r}) \sim p_{pt}(\mathbf{y})$, we can derive that the ground truth R-indices $\mathbf{r} \approx g(f(\bar{\mathbf{x}}, \mathbf{r})) \approx g(h(\bar{\mathbf{x}}, \mathbf{r})) \approx g(\bar{\mathbf{y}})$. Therefore, reconstruction function $g$ can be directly applied to infer R-indices of disease patterns, $\mathbf{r}$, of any unseen PT data.

## 3 IMPLEMENTATION DETAILS

### 3.1 MODEL ARCHITECTURE

The general architecture of networks can be understood from Fig. 1c. The transformation function $f$ utilizes a encoding-decoding structure. Latent variable $\mathbf{z}$ and CN data $\mathbf{x}$ are first mapped to two vectors with dimension 34 and the element-wise multiplication of them are then decoded to construct the synthesized PT data with dimension 139. The inverse function $g$ first decodes real/synthesized PT data $\mathbf{y}/\mathbf{y}'$ to larger vector with dimension 139*M and then maps them into a vector with dimension M. The discriminator $D$ utilizes the encoding structure which maps $y/y'$ to a vector with dimension 2. More details of model architectures can be found in Appendix Table 2 and 3.

### 3.2 LIPSCHITZ CONTINUITY

In implementation, we performed weight clipping to ensure Lipschitz continuity of transformation function $f$ and inverse function $g$ (Arjovsky et al. (2017)). With $\Theta_f$ and $\Theta_g$ denoting the weight spaces of function $f$ and $g$, the compactness of them imply the Lispchitz continuity of two functions, with Lipschitz constants $K_1$ and $K_2$ only depends on $\Theta_f$ and $\Theta_g$. The compactness of $\Theta_f$ and $\Theta_g$ was guaranteed by clapping weight spaces into two closed box, $\Theta_f = [-c_f, c_f]^d$ and $\Theta_g = [-c_g, c_g]^d$. In implementation, both $c_f$ and $c_g$ were set to be 0.5 and these bounds can be further relaxed. Weight clipping was claimed not to be the most satisfactory method to ensure Lipschitz continuity because of difficulty in choosing the appropriate clipping bounds. However, in our case, we performed weight clipping for function $f$ and $g$ rather than discriminator $D$ for a different purpose and weight clipping contributed to good performances for deriving representations.

### 3.3 TRAINING DETAILS

Regarding optimization procedure, ADAM optimizer(Kingma & Ba (2014)) was used with a learning rate (lr) $4 * 10^{-5}$ for Discriminator and $2 * 10^{-4}$ for transformation function $f$ and clustering function $g$. $\beta_1$ and $\beta_2$ are set to be 0.5 and 0.999 respectively. For hyper-parameters, we set $\gamma = 6$, $\kappa = 80$, $\zeta = 80$, $\mu = 500$, $\eta = 6$. However, performance of model on different tasks are robust to varying hyper-parameters as shown in section B.2. Parameter $\lambda$ determines the relative importance of orthogonality loss and controls the degree of overlapping among synthesized patterns. Without prior knowledge on overlapping degree of underlying disease patterns, we set $\lambda$ to different values in the following experiments and used agreements among multiple trained models to determine the optimal one. Moreover, for all experiments, the batch size was set to be 1/8 of the PT data sample sizes. The model was trained for at least 100000 iterations and saved until the reconstruction loss and the monotonicity loss are smaller than 0.003 and $6 * 10^{-4}$ respectively. Higher lr might improve convergence speed in some cases. Detailed training procedure is revealed by Appendix Algorithm1.

## 4 EXPERIMENTS

### 4.1 DATA PREPROCESSING

For semi-synthetic and real data experiments, we used T1-weighted (T1w) MR imaging (MRI) and cognition from the iSAGING (Imaging-based coordinate SysTem for AGIng and NeurodeGenerative diseases) consortium for cognitive impairment and dementia. All baseline MRIs were corrected for intensity inhomogeneities (Sled et al. (1998)). Then, 139 regions of interest (ROIs) of brain volumes were extracted using a multi-atlas label fusion method (Doshi et al. (2015)), harmonized (Pomponio et al. (2020)) to remove site effect, and then utilized as input features for Surreal-GAN. (Fig. 6)

## 4.2 SEMI-SYNTHETIC EXPERIMENTS

**Semi-synthetic data construction** To set the ground truth of disease patterns and severity a prior, we generated semi-synthetic data by imposing simulated disease-related patterns to real CN data, thereby retaining the variation from the real CN data. Specifically, we split 1392 cognitively normal (age < 70 years) into one real CN group with 492 subjects and one Pseudo-PT group with 900 subjects. For the $i_{th}$ Pseudo-PT subject, we sampled a three-dimensional vector as pattern severity from a multivariate uniform distribution, $s_i \sim U[0, 1]^3$. Three types of atrophy patterns were introduced to 900 Pseudo-PT subjects. Different regions were included in different patterns as shown in Appendix Table 4. Five different datasets were constructed for testing the robustness of model under different conditions: (1) **Basic dataset**: For the $i_{th}$ Pseudo-PT subject and the $j_{th}$ ROI included in pattern $k$, we decreased the volume $v_{ij}$ depending on sampled severity: $v_{ij} = v_{ij} - v_{ij} * s_{ik} * N(1, 0.05) * 0.3$. Each pattern consists of changes in 14 ROIs and has four of them shared with other patterns; (2) **Large overlapping dataset**: Same as the Basic dataset, except that each pattern shares 8 ROIs with others; (3) **Scarce regions dataset**: Each pattern consists of changes in only 4 ROIs with other parts the same as the Basic dataset; (4)**Within-pattern noise dataset**: Regional changes incorporated in the same pattern have larger within-pattern variances with $v_{ij} = v_{ij} - v_{ij} * s_{ik} * N(1, \mathbf{0.2}) * 0.3$; (5) **Mild atrophy dataset**: Compared to the basic case, smaller changes were imposed by letting $v_{ij} = v_{ij} - v_{ij} * s_{ik} * N(1, 0.05) * \mathbf{0.2}$.

**Evaluation and Agreement Metric** In our experiments, Concordance Index (c-index) was used as the evaluation metric. Specifically, for each pattern, we calculated a c-index between the inferred R-index and the ground truth. The average of M derived C-indices, refereed as pattern-c-index, was used for model evaluations. Moreover, to quantify the pair-wise agreement between two trained models, we proposed another Pattern-agr-index which equals pattern-c-index between R-indices derived by two different models.

**Experiment Procedure** We first validated the relationship between model agreements and their accuracy. On each dataset, we ran the model 10 times with parameter $\lambda = 0.1, 0.2, 0.4, 0.6, 0.8$ respectively. With each $\lambda$, the average pair-wise pattern-agr-index among 10 models was compared with their average pattern-c-indices. Further, the $\lambda$ value leading to the best agreement was considered as the best choice, and results from 10 corresponding models were reported as model performances for each task. Moreover, we fixed the optimal $\lambda$ value for each task and changed other parameters to 0 to understand contributions from each regularization term (section B.3). Lastly, we compared our model with five other models. NMF and Factor Analysis (FA) are well-known for deriving lower-dimensional representation of complex dataset. The variant opNMF (Sotiras et al. (2014)) has shown promise in parsing complex brain imaging data with extra orthogonal constraint. Discriminant-NMF (DNMF) (Lee et al. (2012)) was proposed to learn representations which best classify subgroups. LDA, though known for extracting topics from documents, have been recently applied to uncover heterogeneity of AD disease from neuroimaging data (Zhang et al. (2016)). Data preprocessing for these models were introduced in section B.4.

## 4.3 REAL DATA EXPERIMENTS

**Data Selection** We applied Surreal-GAN to an AD dataset. For AD, we defined the CN group (N=850) to be subjects with Mini-mental state examination (MMSE) scores above 29, and the PT group (N=2204) as subjects diagnosed as mild cognitive impairment (MCI) or AD at baseline.

**Experiments Procedure** All CN and PT subjects were first residualized to rule out the covariate effects estimated in the CN group using linear regression. Then, adjusted features were standardized with respect to CN group. Without ground truth, we selected both the optimal number of patterns, $M$, and $\lambda$ parameter by measuring agreements among repetitively trained models. For each combination of $M = 2, 3, 4$ and $\lambda = 0.1, 0.2, 0.4, 0.6, 0.8$, we ran the model 10 times respectively. The $M$ and $\lambda$ values leading to the highest agreement was considered optimal. Among the 10 corresponding models, the one having the highest mean pair-wise agreements with the other models was used to derive R-indices for all PT subjects. To visualize patterns corresponding to the $i_{th}$ dimension of R-indices, $\mathbf{r}_i$, we grouped subjects into three different subgroups: $\mathbf{r}_i < 0.4$, $0.4 < \mathbf{r}_i < 0.7$ and $\mathbf{r}_i > 0.7$, with $\mathbf{r}_j < 0.4$ for all $j \neq i$. Voxel-wise group comparisons were performed between each subgroup and the CN group via AFNI-3dttest (R.W. (1996)) and GM-tissue map (Davatzikos et al. (2002)) which encodes the volumetric changes in GM observed during the registration. Pattern representations, R-indices, of PT subjects were further compared with other clinical variables to test their significance. Details of clinical variables and statistical tests can be found in section B.9.

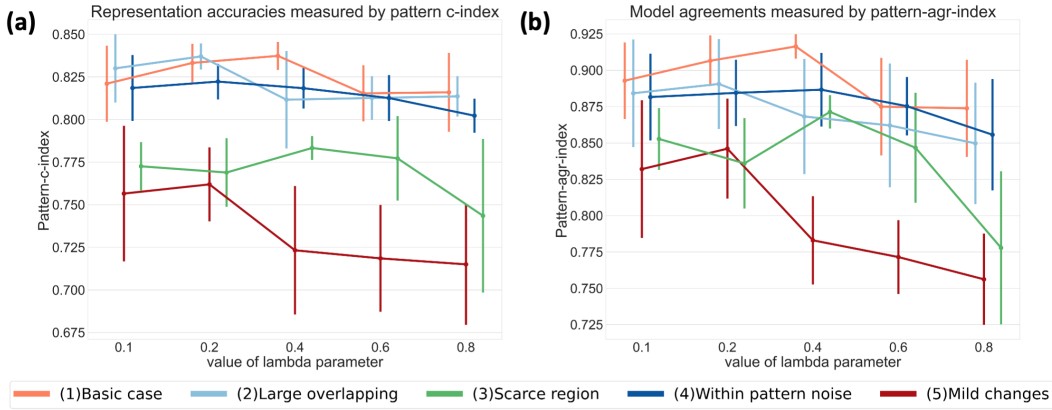

Figure 2: Results on Semi-synthetic data (results presented as mean value ± standard error). (a) Representation accuracies on different tasks (measured by Pattern-c-index); (b) Agreements among repetitively trained models on different tasks (measured by Pattern-agr-index).

Table 1: Model Comparison Results

|  | Surreal-GAN | NMF | LDA | FA | OpNMF | DNMF |
|---|---|---|---|---|---|---|
| Basic Case | **0.838** **± 0.011** | 0.680 ± 0.000 | 0.605 ± 0.001 | 0.631 ± 0.000 | 0.648 ± 0.000 | 0.562 ± 0.017 |
| Large Overlapping | **0.837** **± 0.008** | 0.692 ± 0.000 | 0.591 ± 0.000 | 0.612 ± 0.000 | 0.670 ± 0.000 | 0.566 ± 0.022 |
| Scarce Regions | **0.783** **± 0.007** | 0.625 ± 0.000 | 0.582 ± 0.001 | 0.526 ± 0.000 | 0.608 ± 0.000 | 0.525 ± 0.006 |
| Within-pattern Noise | **0.818** **± 0.013** | 0.675 ± 0.000 | 0.626 ± 0.000 | 0.619 ± 0.000 | 0.651 ± 0.000 | 0.550 ± 0.013 |
| Mild Changes | **0.762** **± 0.023** | 0.611 ± 0.000 | 0.574 ± 0.004 | 0.526 ± 0.000 | 0.594 ± 0.000 | 0.539 ± 0.013 |

## 5 RESULTS

### 5.1 RESULTS ON SEMI-SYNTHETIC DATA

We first validated that agreements among multiple trained models (Fig. 2b) are good indicators of model accuracy (Fig. 2a) for parameter selections. Besides, from Fig. 2a, we proved the model's ability in capturing ground truth of patterns and severity under different cases including pattern overlapping and large within-pattern noise. However, the model experienced declining performances when each pattern includes only scarce disease-related regions, and showed even worse performances when very mild atrophies are led by disease process. This is expected since, for the mild atrophy case, the simulated atrophy rate, 0-20% of the regional volume, is even smaller than the standard deviation of most regional volumes among the normal population. Finally, Surreal-GAN significantly outperformed all five methods on five data sets we constructed (Table 1). However, none of these methods optimally utilize the CN data as a reference distribution, which is one of important limitations preventing them from precisely learning disease-specific pattern representations rather than capturing disease-unrelated confounding variations. The family of DNMF methods capture discriminant representation by minimizing and maximizing within-class and among-class variations respectively, thus did not capture heterogeneity within the PT class accurately either. Designed for parsing disease heterogeneity, semi-supervised methods focused on clustering patients into hard categorical subtypes (Varol et al. (2016), Dong et al. (2015), Wen et al. (2021), Yang et al. (2021)). With different end points, these methods (including Smile-GAN) can not be directly compared with Surreal-GAN. To the best of our knowledge, our model is the first semi-supervised approach for learning disease-related representations, using a healthy reference group as a means for comparison. Therefore, to better understand the model performance, we leveraged the simulated

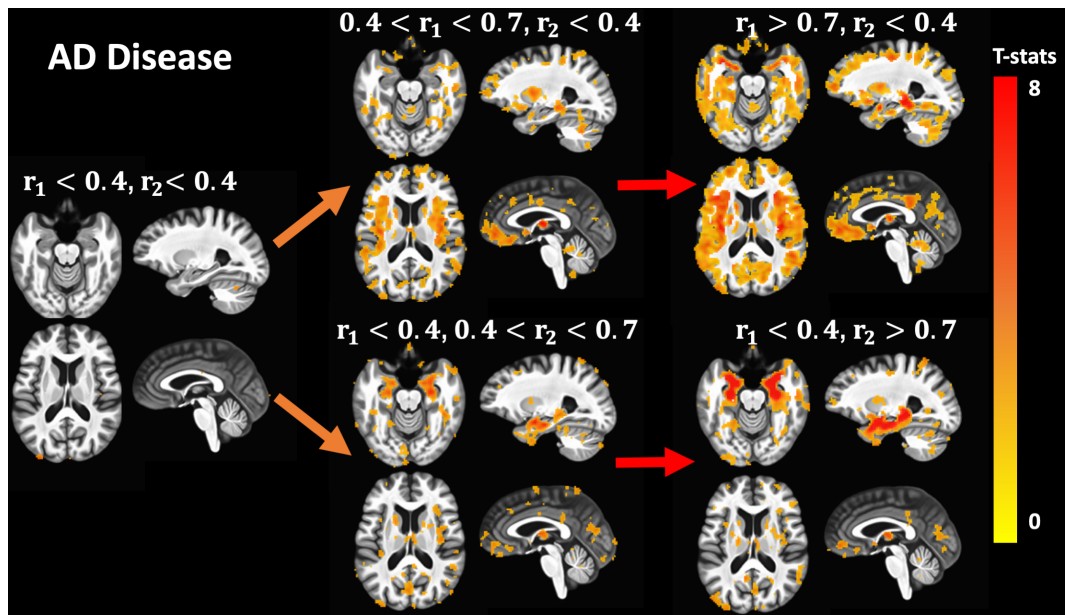

Figure 3: Voxel-wise statistical comparison results between selected subgroups and CN. (FDR method with p-value threshold of 0.05 was used for multiple comparison). In each direction guided by arrows, we increased values of one R-index and keep the other relatively fixed. Therefore, we chose out groups of subjects who have much stronger expression of one pattern for visualization.

ground truth information (not available in real experiments), derived "practical upper bounds" for compared unsupervised models, and also computed the gap between Surreal-GAN and supervised methods. (section B.5 and B.6)

## 5.2 RESULTS ON REAL DATA

On the AD dataset, models show highest agreement when the number of patterns $M = 2$. From voxel-based comparison results (Fig. 3), we can clearly visualize two patterns related to AD. Each dimension of R-indices, $r_i$, shows a clear positive correlation with the corresponding pattern severity. Pattern1, correlated with $r_1$, shows diffuse cortical atrophies (e.g., bilateral insula, orbitofrontal and frontal poles), while pattern2 reveals focal atrophy in medial temporal lobes. Reproducibility of these two patterns were validated in section B.7. Furthermore, the 2D R-indices were effective and interpretable signatures for disease diagnosis and prognosis, achieving similar performance in AD classification compared to original 139 ROIs (section B.8).

Continuity and monotonicity of inferred R-indices further enable us to easily test correlation among them and other clinical variables (Table 7). Both $r_1$ and $r_2$ are highly correlated with CSF-Abeta value and memory dysfunction measured by ADNI-MEM. However, $r_1$ shows stronger correlations with white matter lesion volumes, presence of hypertension, executive function and language functioning, while $r_2$ is more correlated with CSF-Tau, CSF-pTau and presence of APOE-E4 alleles.

## 6 CONCLUSION

In this study, we have proposed a novel method, Surreal-GAN, for learning representations of underlying disease-related imaging patterns. This model has overcome limitations in previous published semi-supervised clustering methods and shown great performance on semi-synthetic data sets. On AD dataset, the model parse heterogeneous PT data into two concise and clinically informative indices with preserved discriminant power, further proving its huge potential in uncovering disease heterogeneity from MRI data.

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

## A  APPENDIX: METHODS

### A.1  LEMMA1

Let $f : \mathbb{X} * \mathbb{Z} \rightarrow \mathbb{Y}$ and $g : \mathbb{Y} \rightarrow \mathbb{X}$ be the transformation and inverse mapping function respectively. If $g$ is K-Lipschitz continuous and $d(\cdot, \cdot)$ is any distance metric satisfying triangular inequality, then for $\forall z_1, z_2 \in \mathbb{Z}$, and $\forall x \in \mathbb{X}$, $d(f(x, z_1), f(x, z_2))$ is lower bounded by $\frac{d(z_1, z_2)}{K} - \frac{1}{K}(d(g(f(x, z_1)), z_1) + d(g(f(x, z_2)), z_2))$

**Proof:** By triangular inequality, we can derive that:

$$d(f(x, z_1), f(x, z_2)) \geq \frac{1}{K} d(g(f(x, z_1)), g(f(x, z_2))) \tag{13}$$

$$\geq \frac{1}{K}(d(z_1, z_2) - d(g(f(x, z_1)), z_1) - d(g(f(x, z_2)), z_2)) \tag{14}$$

$$= \frac{d(z_1, z_2)}{K} - \frac{1}{K}(d(g(f(x, z_1)), z_1) + d(g(f(x, z_2)), z_2)) \tag{15}$$

### A.2  REMARK1

By minimizing the reconstruction loss $L_{\text{recons}}$ defined in section 2.2.2 and 2.2.3, we are maximizing a lower bound of the mutual information between latent variable $\mathbf{z}$ and synthesized PT data $\mathbf{y}' = f(\mathbf{x}, \mathbf{z})$. Considering $Q(\mathbf{z}|\mathbf{y}')$ to be an approximation of distribution $P(\mathbf{z}|f(\mathbf{x}, \mathbf{z}))$ and assuming that it follows a Gaussian distribution $\mathrm{N}(g(\mathbf{y}'), \Sigma)$, mutual information denoted by $I$ and entropy denoted by $H$, we can derive that:

$$I(\mathbf{z}, f(\mathbf{x}, \mathbf{z})) = H(\mathbf{z}) - H(\mathbf{z}|f(\mathbf{x}, \mathbf{z})) \tag{16}$$

$$= \mathbb{E}_{\mathbf{y}' \sim p_{syn}(\mathbf{y}')}[\mathbb{E}_{\mathbf{z}' \sim p(\mathbf{z}|\mathbf{y}')}[\log P(\mathbf{z}'|\mathbf{y}')]] + H(z) \tag{17}$$

$$= \mathbb{E}_{\mathbf{y}' \sim p_{syn}(\mathbf{y}')}[D_{\text{KL}}(P(\cdot|\mathbf{y}')||Q(\cdot|\mathbf{y}')) + \mathbb{E}_{\mathbf{z}' \sim p(\mathbf{z}|\mathbf{y}')}[\log Q(\mathbf{z}'|\mathbf{y}')]] + H(\mathbf{z}) \tag{18}$$

$$\geq \mathbb{E}_{\mathbf{y}' \sim p_{syn}(\mathbf{y}')}[\mathbb{E}_{\mathbf{z}' \sim p(\mathbf{z}|\mathbf{y}')}[\log Q(\mathbf{z}'|\mathbf{y}')]] \tag{19}$$

$$= \mathbb{E}_{\mathbf{z} \sim p_{lat}(z), \mathbf{y}' \sim p_{syn}(f(\mathbf{x}, \mathbf{z}))}[\log Q(\mathbf{z}|\mathbf{y}')] \tag{20}$$

$$= \mathbb{E}_{\mathbf{z} \sim p_{lat}(z), \mathbf{x} \sim p_{cn}(\mathbf{x})}[[\log Q(\mathbf{z}|f(\mathbf{x}, \mathbf{z}))] \tag{21}$$

$$\approx \frac{1}{nm} \sum_{i=1}^{n} \sum_{j=1}^{m} (-\frac{\ln|\Sigma|}{2} - \frac{1}{2}(\mathbf{z}_i - f(\mathbf{x}_j, \mathbf{z}_i))^T \Sigma^{-1}(\mathbf{z}_i - f(\mathbf{x}_j, \mathbf{z}_i)) + \text{constant}) \tag{22}$$

(21) and (22) follows the Lemma 5.1 in Info-GAN (Chen et al. (2016)) and law of the unconscious statistician (Lotus) Theorem respectively. Therefore, we derive that the mutual information between the latent variable $\mathbf{z}$ and synthesized PT data $\mathbf{y}'$ is bounded below by $\frac{1}{nm} \sum_{i=1}^{n} \sum_{j=1}^{m} (-\frac{\ln|\Sigma|}{2} - \frac{1}{2}(\mathbf{z}_i - f(\mathbf{x}_j, \mathbf{z}_i))^T \Sigma^{-1}(\mathbf{z}_i - f(\mathbf{x}_j, \mathbf{z}_i)) + \text{constant})$. With a further assumption that $\Sigma$ is an identity matrix, maximization of this lower bound is equivalent to minimizing the reconstruction loss $L_{\text{recons}}$.

### A.3  NETWORK ARCHITECTURE

Table 2 and Table 3 display network architectures of transformation function $f$, discriminator $D$ and reconstruction function $g$ (consisting of $g_1$ and $g_2$).

Table 2: Architecture of transformation function $f$

|  | Layer | Input Size | Bias Term | leaky relu $\alpha$ | Output Size |
|---|---|---|---|---|---|
| Encoder from $\mathbf{x}$ | Linear1+Leaky-Relu | 139*1 | No | 0.2 | 69*1 |
|  | Linear2+Leaky-Relu | 69*1 | No | 0.2 | 34*1 |
| Decoder from $\mathbf{z}$ | Linear1+Sigmoid | M*1 | Yes | NA | 34*1 |
| Decoder to $\mathbf{y}'$ | Linear1+Leaky-Relu | 34*1 | No | 0.2 | 69*1 |
|  | Linear2+Leaky-Relu | 69*1 | No | 0.2 | 139*1 |

Table 3: Architecture of discriminator $D$, Decomposer $g_1$ and Reconstruction $g_2$

|  | Layer | Input Size | Bias Term | leaky relu $\alpha$ | Output Size |
|---|---|---|---|---|---|
| Discriminator | Linear1+Leaky-Relu | 139*1 | Yes | 0.2 | 69*1 |
|  | Linear2+Leaky-Relu | 69*1 | Yes | 0.2 | 34*1 |
|  | Linear3+Softmax | 34*1 | Yes | NA | 2*1 |
| $g_1$ | Leaky-Relu+Linear1 | 139*1 | Yes | 0.2 | (139*M)*1 |
| $g_2$ | Linear1+Leaky-Relu | 139*1 | Yes | 0.2 | 69*1 |
|  | Linear2+Leaky-Relu | 69*1 | Yes | 0.2 | 34*1 |
|  | Linear3+Softmax | 34*1 | Yes | NA | M*1 |

### A.4 ALGORITHM

Detailed training procedure of Surreal-GAN is disclosed by Algorithm 1.

**Algorithm 1:** Surreal-GAN training procedure. $l_c$ represents cross entropy loss and $e_i$ represents a one hot vector with 1 at $i_{th}$ component. $\hat{q}_{f(x,z)}$, $\hat{l}_{g_2(g_1(f(x,z)))}$ and $A_{f(x,z)}$ are defined in section 2.2.3 and 2.2.4.

**while** *not meeting stopping criteria or reaching max_epoch* **do**

    **for** *all batches* $\{x_i\}_{i=1}^m, \{y_i\}_{i=1}^m$ **do**

        sample m latent vectors $\{z_i\}_{i=1}^m$ from multivariate uniform distribution with $z_i \sim U[0,1]^M$

        **Update weights of discriminator $D$:** Use ADAM to update $\theta_D$ with gradient:
$\nabla_{\theta_D} \frac{1}{m} \sum_{i=i}^{m}[(l_c(D(y_i),e_1) + l_c(D(f(x_i,z_i),e_0)))]$

        sample m latent vectors $\{z_i'\}_{i=1}^m$ conditioned on previously sampled $\{z_i\}_{i=1}^m$, with each $z_{i_j}' \sim U(z_{i_j},1]^M$
sample another m latent vectors $\{z_i^{cn}\}_{i=1}^m$ with each $z_i^{cn} \sim U(0,0.05)^M$

        **Update weights of transformation function $f$:** Use ADAM to update $\theta_f$ with gradient: $\nabla_{\theta_f} \frac{1}{m} \sum_{i=i}^{m}[l_c(D(f(x_i,z_i),e_1))) + \gamma(||f(x_i,z_i) - x_i||_1) + \kappa||g_1(f(x_i,z_i)) - \hat{q}_{f(x_i,z_i)}||_2 + \zeta||\hat{l}_{g_2(g_1(f(x_i,z_i)))} - z_i||_2 + \lambda||A_{f(x_i,z_i)}^T A_{f(x_i,z_i)} - I||_F + \mu||\max(|f(x_i,z_i) - x_i| - |f(x_i,z_i') - x_i|, \mathbf{0})||_2 + \eta||f(x_i,z_i^{cn}) - x_i||_1]$

        **Update weights of function $g_1$:** Use ADAM to update $\theta_{g_1}$ with gradient:
$\nabla_{\theta_{g_1}} \frac{1}{m} \sum_{i=i}^{m}[||g_1(f(x_i,z_i)) - \hat{q}_{f(x_i,z_i)}||_2]$

        **Update weights of function $g_2$:** Use ADAM to update $\theta_{g_2}$ with gradient:
$\nabla_{\theta_{g_2}} \frac{1}{m} \sum_{i=i}^{m}[||\hat{l}_{g_2(g_1(f(x_i,z_i)))} - z_i||_2$

    **end**

**end**

## B APPENDIX: EXPERIMENTS

### B.1 REGIONS WITH SIMULATED ATROPHY IN SEMI-SYNTHETIC DATASETS

Regions of interest (ROI) included in three patterns (P1-P3) in different semi-synthetic datasets are revealed by Table 4. Small overlapping case indicates the included ROIs in three different dataset as described in section 4.2: (1) Basic dataset; (4)Within-pattern noise dataset and (5)Mild atrophy dataset. Each ROI is a combination of left and right parts of the region, so each check sign represents inclusion of 2 out of 139 regions.

Table 4: ROIs included in synthetic patterns (OP:opercular part; TP: triangular part)

| ROI | Small overlapping | | | Large overlapping | | | Scarce Region | | |
|---|---|---|---|---|---|---|---|---|---|
| | P1 | P2 | P3 | P1 | P2 | P3 | P1 | P2 | P3 |
| Amygdala | ✓ | | | ✓ | ✓ | | ✓ | | |
| Hippocampus | ✓ | ✓ | | ✓ | ✓ | | ✓ | | |
| Angular gyrus | | | ✓ | | | ✓ | | | ✓ |
| entorhinal area | ✓ | | | ✓ | | | | | |
| frontal operculum | | ✓ | | | ✓ | | | | |
| inferior temporal gyrus | ✓ | | | ✓ | | | | | |
| lateral orbital gyrus | | | ✓ | | | ✓ | | | |
| medial frontal cortex | | ✓ | ✓ | | ✓ | ✓ | | | |
| middle frontal gyrus | | ✓ | | | ✓ | ✓ | | ✓ | |
| middle occipital gyrus | | | ✓ | | | ✓ | | | |
| OP of the inferior frontal gyrus | | ✓ | | | ✓ | | | ✓ | |
| parahippocampal gyrus | ✓ | | | ✓ | | | | | |
| posterior insula | | ✓ | | | ✓ | | | | |
| parietal operculum | ✓ | | ✓ | ✓ | | ✓ | | | |
| supramarginal gyrus | | | ✓ | | | ✓ | | | |
| superior parietal lobule | | | ✓ | ✓ | | ✓ | | | ✓ |
| temporal pole | ✓ | | | ✓ | | | | | |
| TP of the inferior frontal gyrus | | ✓ | | | ✓ | | | | |

## B.2 HYPER-PARAMETER SELECTION

We trained the model on the basic case and the mild changes case (the easy and hard case) with varying hyper-parameters to test the model robustness to hyper-parameter selections. Specifically, we changed each parameter from 50% to 150% of the preset value (introduced in section 3.3) while keeping other parameters fixed. With each set of parameters, we trained the model 10 times and reported average pattern-c-index and average pattern-agr-index as described in section 4.2. From Fig. 4a, we can see that model performances are mostly robust to different choices. For the mild changes case, models, with different $\eta$ values, do show higher variations in performances. However, pattern-agr-indices still have good indication of the best choice, and this metric can always be used for parameter selections on different datasets.

## B.3 CONTRIBUTION FROM REGULARIZATION

Similar to section B.2, we set different parameters to 0 to understand contributions from corresponding regularization terms. Specifically, we set $\kappa = 0$ (no decomposition loss), $\lambda = 0$ (no orthogonality loss), $\mu = 0$ (no monotonicity loss), $\zeta = 0$ (no cn loss), $\mu/\zeta = 0$ (no constraint on monotonicity and positive correlation), $\kappa/\lambda/\mu/\zeta = 0$ (no extra regularization compared to Smile-GAN framework) respectively with other parameters fixed. With different sets of parameter values, we trained the model 10 times and report average pattern-c-indices. From Fig.4c, we can clearly understand the importance of different regularization terms and visualize how additional regularization terms lead to improved representation performances compared to a pseudo comparable "Smile-GAN" (Using same regularization as Smile-GAN but with continuous latent variable).

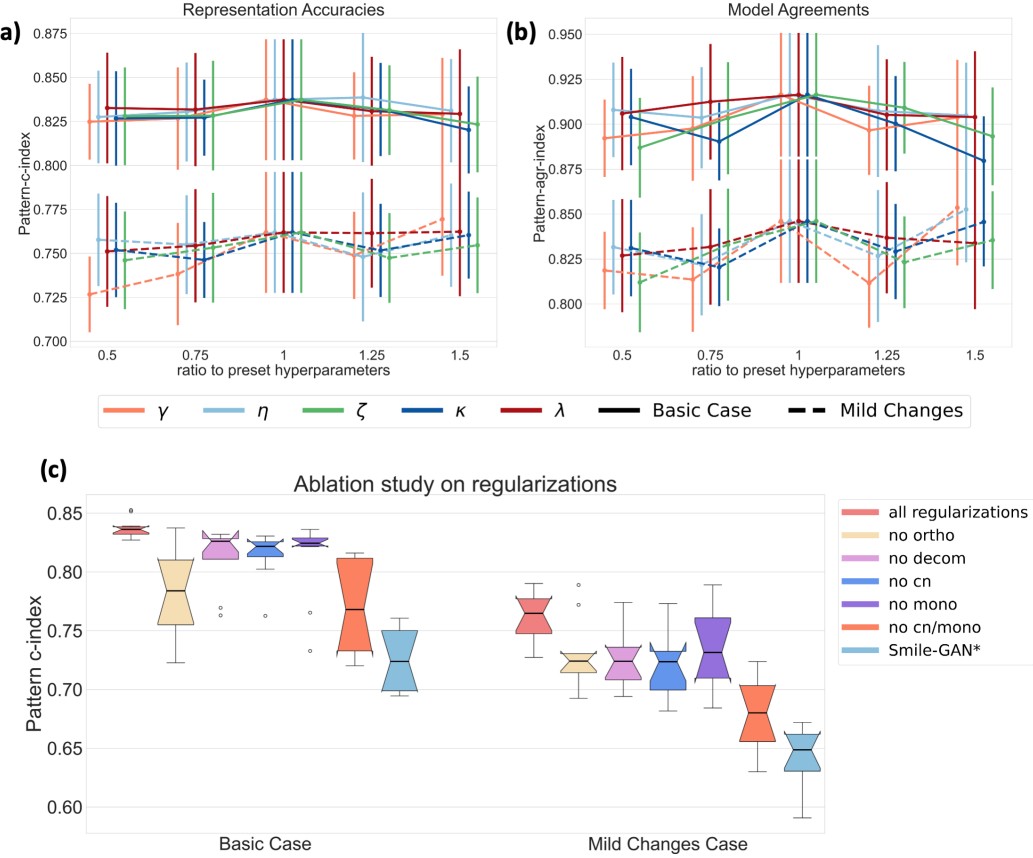

Figure 4: Regularization and hyper-parameters. (a) Representation accuracy on different tasks with variations in hyper-parameters (measured by Pattern-c-index); (b) Agreements among repetitively trained models on different tasks with variations in hyper-parameters (measured by Pattern-agr-index) (results presented as mean value ± standard error for (a,b)); (c) Representation accuracy with regularization terms removed. (*Smile-GAN here refers to a "pseudo Smile-GAN" with the same regularization terms included in Smile-GAN model. Latent variable $z$ is sampled from same continuous distribution as in Surreal-GAN.)

Table 5: "Practical upper bounds" of compared models

|  | Surreal-GAN | NMF | LDA | FA | OpNMF | DNMF |
|---|---|---|---|---|---|---|
| Basic Case | 0.838 ± 0.011 | 0.834 ± 0.000 | 0.769 ± 0.005 | 0.750 ± 0.000 | 0.751 ± 0.000 | 0.631 ± 0.017 |
| Large Overlapping | 0.837 ± 0.008 | 0.844 ± 0.000 | 0.777 ± 0.001 | 0.721 ± 0.005 | 0.715 ± 0.000 | 0.641 ± 0.017 |
| Scarce Regions | 0.783 ± 0.007 | 0.760 ± 0.000 | 0.640 ± 0.015 | 0.618 ± 0.001 | 0.666 ± 0.000 | 0.548 ± 0.006 |
| Within-pattern Noise | 0.818 ± 0.013 | 0.831 ± 0.000 | 0.775 ± 0.003 | 0.764 ± 0.000 | 0.730 ± 0.015 | 0.644 ± 0.031 |
| Mild Changes | 0.762 ± 0.023 | 0.756 ± 0.000 | 0.657 ± 0.028 | 0.613 ± 0.000 | 0.641 ± 0.000 | 0.571 ± 0.015 |

## B.4 DATA PREPROCESSING FOR COMPARED MODELS

Z-scores of ROIs with respect to the CN group were directly used as input features of FA model. However, for NMF, OpNMF and LDA model, we need to borrow a prior knowledge that atrophy (volume decreasing) is more common in dementia-related brain diseases, and thus, reversed the sign

Table 6: Comparisons with supervised regression models

| Task | Surreal-GAN | | LR | | SVR | |
|---|---|---|---|---|---|---|
| | **Train** | **Test** | **Train** | **Test** | **Train** | **Test** |
| **Basic Case** | 0.827 ±0.005 | 0.830 ±0.007 | 0.896 ±0.001 | 0.871 ±0.003 | 0.926 ±0.001 | 0.863 ±0.006 |
| **Large Overlapping** | 0.828 ±0.008 | 0.827 ±0.007 | 0.901 ±0.001 | 0.876 ±0.003 | 0.927 ±0.001 | 0.869 ±0.003 |
| **Sparse Regions** | 0.764 ±0.010 | 0.761 ±0.014 | 0.853 ±0.001 | 0.816 ±0.005 | 0.932 ±0.002 | 0.804 ±0.009 |
| **Within Pattern Noise** | 0.813 ±0.007 | 0.812 ±0.009 | 0.890 ±0.001 | 0.862 ±0.003 | 0.926 ±0.001 | 0.857 ±0.009 |
| **Mild Changes** | 0.741 ±0.010 | 0.743 ±0.017 | 0.853 ±0.001 | 0.817 ±0.003 | 0.929 ±0.009 | 0.811 ±0.005 |

of z-scores, kept positive z-scores, and set negative values to 0. For LDA model, we further discretized the value by first multiplying z-scores by 10 and then mapping them to the closest integer as described in Zhang et al. (2016). For the DNMF model, both CN and PT data and their corresponding diagnosis labels are provided as input, with ROI data first standardized to lie in range 0-1.

## B.5 "PRACTICAL UPPER BOUNDS" OF UNSUPERVISED METHODS

As discussed in section 5.1, unsupervised methods were not designed to best utilize the CN data as a reference distribution. Thus, we derived "practical upper bounds" for compared models on different tasks, with the assumption that these methods have some components capturing the ground truth while having other components representing non-disease related variations. On each semi-synthetic test, we ran each method with number of components, $M$, ranging from 3-10. For $M > 3$, we intensively searched for the combination of 3 components that best match the ground truth and calculated the pattern c-index. The best result among $M = 3 - 10$ is considered as an "practical upper bound" for the model. As shown in Table 5, upper bounds of NMF methods are comparable to results from Surreal-GAN, while other methods still show much worse performances. Note that these are "practical upper bounds" derived with known ground truth. In real cases, we are not interested in a set patterns that includes the true disease effects but want to find disease-related patterns only. However, it will be very hard to know what the best number of components is and which components are truly disease-related with these compared models. In contrast, Surreal-GAN can directly and accurately output disease-specific representations without indices capturing non-disease related variations, which is one of the key advantages of the proposed method.

## B.6 COMPARISONS WITH SUPERVISED METHODS

We further compared the model with linear regression (LR) and support vector regression (SVR) with RBF kernel to understand the gap between our model and the most optimal model performances in semi-synthetic tests. A five fold cross validation were run with three different models. Average pattern-c-indices on both train and test sets are reported. We can first observe that Surreal-GAN, as an semi-supervised methods, shows almost equal performances on train and test set, proving a good ability in generalization. On five different tasks (table6), there is a 0.04-0.07 pattern-c-indices gap between Surreal-GAN and supervised model performances. This gap is considered reasonable but there is still space for model improvements.

## B.7 GENERALIZATION AND REPRODUCIBILITY ON REAL DATASET

We randomly half split the real data (described in section 4.3) in train and test set, with both CN and PT data equally distributed in two sets. Following same experiment procedure described in section 4.3, we first trained the model on the training set, and then applied the model to derive 2-dimensional R-indices for all CN and PT data in the test set. Voxel-wise group comparisons were

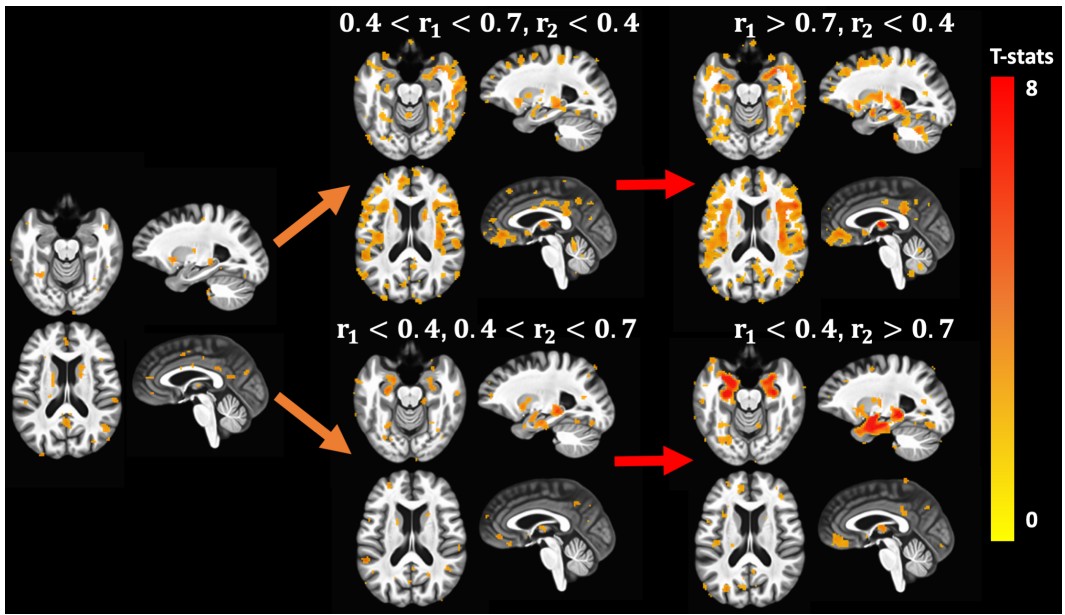

Figure 5: Voxel-wise statistical comparison results on test set. (FDR method with p-value threshold of 0.05 was used for multiple comparison).

conducted between subgroups and CN in the test set for visualization. From Fig. 5, we can observe that R-indices are correlated with almost same patterns shown in Fig. 3, though much smaller sample size led to relatively lower statistical significance.

## B.8   LOW DIMENSIONAL REPRESENTATION FOR AD DIAGNOSIS

We used R-indices of test data introduced in section B.7, and compared their ability to classify CN and PT with that of 139 ROI volumes. Support vector machine (SVM) was selected as the classification model. $20\%$ leave-out experiments were performed 100 times with derived R-indices and 139 ROI as input features respectively. In each run, we randomly sampled $80\%$ of data as training set and applied trained models on the rest $20\%$ to derive the AUC metric. Grid search were used for parameter selection. With 2D R-indices as features, SVM model was able to reach an average AUC, $0.895 \pm 0.011$, which is only mildly lower than AUC derived with 139 ROIs as features, $0.929 \pm 0.015$. Therefore, we can conclude that, 2D R-indices, kept almost all discriminant information among PT data, while parsing the heterogeneity in a clear, concise and interpretable way.

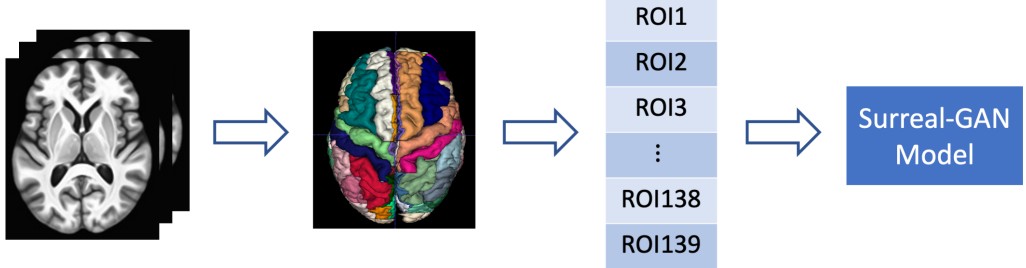

Figure 6: Procedure for deriving Surreal-GAN input features (ROI volumes): MRI images were segmented using the MUSE method (Doshi et al. (2015)). Volumes of segmented regions (ROI volumes) were then extracted as input features for Surreal-GAN model.

### B.9 ASSOCIATIONS WITH CLINICAL VARIABLES

Each dimension of R-indices, $\mathbf{r}_i$, is compared with different clinical variables for AD disease. For each clinical variable, we only selected out samples who have it available for statistical analyses. ADNI composite scores, including ADNI-EF, ADNI-MEM and ADNI-LAN, measure subjects' functions in executive functions, memory and language respectively. They are only available among subjects from Alzheimer's Disease Neuroimaging Initiative (ADNI) study. Digit Symbol Substitution Test (DSST) measures a range of cognition operations including motor speed, attention, and visuoperceptual. For all mentioned clinical scores, a higher value indicates a better performance. CSF-Abeta, CSF-Tau and CSF-PTau are three very important hallmarks of AD disease corresponding to amyloid and tau hypotheses. Since measures of these three values have great variance among different studies, only ADNI samples with CSF measures were included for analysis. For dichotomous variables (Apoe-E4 Alleles Carrier & Hypertension), point biserial correlation with corresponding p value was derived between the variable and each pattern dimension, $\mathbf{r}_i$. Pearson correlations were derived for other continuous variables.

Table 7: Correlation between pattern severity and clinical variables for AD disease

|  | $\mathbf{r}_1$ | $\mathbf{r}_2$ |
|---|---|---|
| White matter lesion volumes | 0.415 (**p<0.001**) | 0.184 (**p<0.001**) |
| DSST | -0.055 (p=0.603) | -0.212 (**p=0.042**) |
| Apoe-E4 Alleles Carrier | 0.003 (p=0.965) | 0.212 (**p<0.001**) |
| Hypertension | 0.181 (**p<0.001**) | -0.006 (p=0.890) |
| CSF-Abeta | -0.238 (**p<0.001**) | -0.302 (**p<0.001**) |
| CSF-Tau | 0.111 ( **p=0.002**) | 0.270 (**p<0.001**) |
| CSF-PTau | 0.059 (p=0.103) | 0.152 (**p<0.001**) |
| ADNI-EF | -0.409 (**p<0.001**) | -0.240 (**p<0.001**) |
| ADNI-MEM | -0.453 (**p<0.001**) | -0.466 (**p<0.001**) |
| ADNI-LAN | -0.422 (**p<0.001**) | -0.281 (**p<0.001**) |

