# OpenReview forum: "Surreal-GAN:Semi-Supervised Representation Learning via GAN for uncovering heterogeneous disease-related imaging patterns"
_ICLR.cc/2022/Conference — ICLR 2022 Poster_

### Official Review · Reviewer_8j3t · 2021-11-01

**Correctness:** 3
**Technical Novelty And Significance:** 3
**Empirical Novelty And Significance:** 2
**Recommendation:** 6
**Confidence:** 4

**Main Review:**

The paper is well clearly organized, and the general idea of the method is very interesting. However, there are a few things I would like to address.
1. It is not easy to understand the actual model input and how it is generated from the structural MR images. An example image of the preformed atlas segmentation would be beneficial for understanding.
2. Figure 1 is difficult to understand. Is Figure 1b the result of the experiments or an illustration of what to expect from the presented method?
3. In the introduction and later in the paper, the authors refer to the low dimensional representation as R indices r_i and refer to Figure 1b. However, Figure 1b only shows z_1 and z_2. It would be great if the authors could comment on the relation between R indices and the latent representation z.
4. For the model architecture, the latent variable and the input are mapped into a 34-dimensional space. According to the paper, the dot product is applied to both vectors. However, the input to the decoder is 34x1. Is an element-wise multiplication used instead of the dot product?
5. It is not clear how the evaluation of the model f for each z_i is done separately for the pattern decomposition. It would be great if the authors could comment on this.
6. It is not clear what the number of features S relates to in the paper.
7. For the evaluation, all necessary information is provided. However, the selection method of the hyperparameter for the loss function is not given. Furthermore, it is not evaluated how the selection of the hyperparameter influences the outcome of the model. This is only done for lambda and M.
8. The y axis label of Figure 2b is cut off.
9. The authors mentioned that this method is an extension of the Smile-GAN. Is there a specific reason why the Smile-GAN is not used for comparison?


**Summary Of The Paper:**

In this paper, the authors present a method for learning a representation of disease-related image patterns from regional volume information generated from structural MRI images. The method is based on a GAN approach, and different regularization methods on the latent representation are presented to improve the learning of disease-related representations. There are four contributions presented in the paper:
1. Modelling a disease as a continuous process
2. Controlling the monotonicity of the disease pattern
3. Separation of captured disease pattern
4. Inverse Mapping to ensure that disease patterns synthesized are captured

The evaluation of the method is performed on semi-synthetic data sets and an actual Alzheimer’s disease data set. The results show that the method can identify two clinically informative patterns.


**Summary Of The Review:**

The presented methods present some interesting ideas for modelling disease-related representation. However, the description of the presented method needs some improvements.

---

> ### Author Response · Authors · 2021-11-20
> **Response to Reviewer 8j3t (Part1)**
>
> 1. It is not easy to understand the actual model input and how it is generated from the structural MR images. An example image of the preformed atlas segmentation would be beneficial for understanding.
>
> Response: Thank the reviewer for pointing out this confusing point. We have included a new figure (Fig. 6) describing the atlas segmentation process.
>
> 2. Figure 1 is difficult to understand. Is Figure 1b the result of the experiments or an illustration of what to expect from the presented method?
>
> Response: We agree that this might be confusing. Figure 1b actually shows the results of Surreal-GAN on AD data. However, Figure 1b aims to help readers visualize differences between Surreal-GAN and clustering methods, and help understand the importance of this work. Thus, we call them expected improvement from Surreal-GAN before talking about details of model and experiments. We have changed our language to make it clearer.
>
> 3. In the introduction and later in the paper, the authors refer to the low dimensional representation as R indices r_i and refer to Figure 1b. However, Figure 1b only shows $z_1$ and $z_2$. It would be great if the authors could comment on the relation between R indices and the latent representation $z$.
>
> Response: We apologize for the confusion led by the figure. Latent variable $z$ is the input for the transformation function $f$ in the training procedure, while low dimensional representation $r$ is the output of the inverse function $g$.  During the training procedure, because of the reconstruction regularization, R indices inferred from synthesized patient data should be very close to the latent variable $z$ used to synthesize it through the transformation function. Therefore, r is actually an approximation of the ground truth $z$ which generates patient data through underlying transformation. After training procedure, inverse function $g$ is used to infer the R indices of any real patient data when ground truth is not available and $r$ is used for any downstream analysis. In figure 1b, we agree that $z_1$ and $z_2$ are confusing, while $r_1$ and $r_2$ are more suitable notations in that context. We have adjusted the figure to make it clearer.
>
> 4. For the model architecture, the latent variable and the input are mapped into a 34-dimensional space. According to the paper, the dot product is applied to both vectors. However, the input to the decoder is 34x1. Is an element-wise multiplication used instead of the dot product?
>
> Response: Thank the reviewer for catching this error. It should be an element-wise multiplication rather than a dot product, and we have corrected this in the manuscript.
>
> 5. It is not clear how the evaluation of the model $f$ for each $z_i$ is done separately for the pattern decomposition. It would be great if the authors could comment on this.
>
> Response: Based on the reviewer’s comment, we think that the reviewer is referring to the decomposition function, $g_1$, introduced in section 2.2.3. We did not evaluate pattern decomposition, since we did not directly use this function in any experiments and, basically, R-indices will be more of interest to downstream clinical analysis. Here pattern decomposition mostly serves for improving accuracies of inferred R-indices. Improvements led by decomposition function can be found in section A.8. However, we did not deny that pattern decomposition might be a good tool for interpreting R-indices and we will evaluate them in downstream analysis.
>
> 6. It is not clear what the number of features S relates to in the paper.
>
> Response: The number of features S is the dimension of input features. In later experiments, S=139 is the number of ROIs. We have made this point clearer in the manuscript.
>
> 7. For the evaluation, all necessary information is provided. However, the selection method of the hyperparameter for the loss function is not given. Furthermore, it is not evaluated how the selection of the hyperparameter influences the outcome of the model. This is only done for lambda and M.
>
> Response: We agree that it is important to study the influence from different hyperparameters. In section A.7, we evaluated the model with varying hyper-parameter values. In conclusion, model performance in different tasks were shown to be robust to parameter values selected from 50%-150% of preset values (introduced in section 3.3). Moreover, since agreements among models is shown to be indicative of representation accuracies, pattern-agr-index can be used as a metric for parameter selection if the user wants to refine hyper-parameters when applying the model to a different dataset.
>
> 8. The y axis label of Figure 2b is cut off.
>
> Response: Thank the reviewer for catching this mistake and we have corrected it.

---

> > ### Author Response · Authors · 2021-11-20
> > **Response to Reviewer 8j3t (Part2)**
> >
> > 9. The authors mentioned that this method is an extension of the Smile-GAN. Is there a specific reason why the Smile-GAN is not used for comparison?
> >
> > Response: Thank the reviewer for this question. As in our response to comment3 from reviewer 1 and comment2 from reviewer 2, Smile-GAN and Surreal-GAN are not directly comparable since they seek different end points, where Smile-GAN, with categorical latent variable, aims to cluster patients into a hard categorical subtype membership, whereas Surral-GAN dissects the heterogeneity into a continuous representation, with both an continuous z variable and various additional but effective regularization terms. Completely different structure of results from two methods make it impossible to directly compare Surreal-GAN with Smile-GAN. With that being said, we also studied the effect of the additional regularization terms by comparing Surreal-GAN with a pseudo comparable “Smile-GAN” (Using continuous latent variable with same regularizations as Smile-GAN model). With all additional regularization terms removed, pseudo comparable “Smile-GAN” shows much worse performance than the Surreal-GAN model (Fig. 4c). Again, we want to emphasize that, even without these regularization terms, the model is still significantly different from the original  Smile-GAN model. To make it clear, we refer to it as the “pseudo Smile-GAN framework” above and in the appendix, and we provided more explanation on differences between Surreal-GAN and Smile-GAN in section 2 and 5.1.

---

> > > ### Comment · Reviewer_8j3t · 2021-11-29
> > > **Comment to author response**
> > >
> > > Thank you very much for clarification, I would increase my score to 6.

---

> > > > ### Author Response · Authors · 2021-11-29
> > > > **Response to Reviewer 8j3t**
> > > >
> > > > Thank you very much!

---

### Official Review · Reviewer_QqEE · 2021-11-02

**Correctness:** 3
**Technical Novelty And Significance:** 3
**Empirical Novelty And Significance:** 2
**Recommendation:** 8
**Confidence:** 3

**Main Review:**

Strong Points

1.	Perhaps the biggest shining point is the astonishing number of carefully crafted regularizations – a total of six regularization terms in a single loss function. They are delineated in significant detail from section 2.2.1 to 2.2.5. I appreciate the amount of thinking and hard work that probably goes behind the scene, and I believe the author did a good job persuading me why each of these regularizations are helpful and what they respectively achieve. I particularly liked how the authors pointed out some potential trivial solutions the model can cheat with if proper regularization were not in place.
2.	As for the results, I would consider the voxel-wise statistical comparisons on the real data a rather definitive result demonstrating that Surreal GAN was able to isolate two major sources/locations of atrophy in Alzheimer’s disease. The two distinctive patterns, one showing up across multiple cortical regions and the other localized to the medial temporal lobe would probably be appreciated by many neurobiologists. Meanwhile, the statistical power is quite strong, given that it has been corrected for multiple comparison using the FDR method.


Weak Points

1.	As many as six regularization terms, while I stated as a “strong point”, can also been viewed as a sign of over-engineering. It will be better rationalized if the authors can conduct an ablation study on these five regularizations. With that being said, I acknowledge it might be an unrealistic commitment depending on how long each experiment would take, so this is a soft suggestion rather than an explicit request.
2.	It appears that the authors are utilizing the gradient clipping method to ensure Lipschitz continuity (section 3.2). However, that method is relatively deprecated, as emphasized in the original Wasserstein GAN paper, “weight clipping is a clearly terrible way to enforce a Lipschitz constraint”. While the authors are aware of the case and are conscious about their different use case, nevertheless, alternative ways such as gradient penalty (Improved Training of Wasserstein GANs by Gulrajani et al 2017, aka., WGAN-GP) may be worth considering.
3.	The baseline that the authors compared against are not very competitive. It might be beneficial if a supervised method (is it possible?) can be included, perhaps to set a “practical upper-bound” so that we can see how far the gap there is between that and the proposed method.


Questions to authors

1.	Section 3.3. When the authors mentioned that “the model was trained for at least 100000 epochs and saved until the reconstruction loss is smaller than 0.003 and the monotonicity loss is smaller than 6E-4”, I was surprised by the huge number of epochs. My primary concerns are: 1) does that mean the initial learning rate is a bit too low and/or a bad learning rate scheduling is used? 2) how do you avoid overfitting, besides using the suite of regularizations in the loss function?
2.	Section 2.3. The full loss formula is indexed twice (14 and 15). Is it a mistake?

**Summary Of The Paper:**

In this paper, the authors pointed out the lack of a good method to predict disease progression in a continuum based on representations explicitly modeling specific disease effects. To tackle this problem, they proposed a novel method called Surreal-GAN, a tailored version of generative adversarial networks, to learn separate representations of neurological and neuropsychiatric diseases in an unsupervised manner. The method frames the diseased brains as multiple disease-related features at various severity imposed on normal brains, and by learning these features and severities, it can generate a representation with improved reliability and explanability.

**Summary Of The Review:**

Overall, I would recommend this paper to be accepted. Personally I find the problem the authors are tackling legitimate and usually neglected, and I am persuaded that the proposed solution has some generalizability.

---

> ### Author Response · Authors · 2021-11-20
> **Response to Reviewer QqEE (Part1)**
>
> 1. As many as six regularization terms, while I stated as a “strong point”, can also been viewed as a sign of over-engineering. It will be better rationalized if the authors can conduct an ablation study on these five regularizations. With that being said, I acknowledge it might be an unrealistic commitment depending on how long each experiment would take, so this is a soft suggestion rather than an explicit request.
>
> Response: Thank the reviewer for this constructive suggestion. Based on the suggestion, we conducted the ablation experiments on the basic case and mild changes case, and showed results in section A.8. All regularization terms contribute to better performances of the model, with contributions being more significant in the harder case. Also, as discussed in our responses to previous reviewers’ comments, with all additional regularization terms removed, a pseudo comparable “Smile-GAN” shows much worse performance than the Surreal-GAN model. This also helps prove that these additionally designed regularizations are indeed important for the Surreal-GAN model.
>
> 2. It appears that the authors are utilizing the gradient clipping method to ensure Lipschitz continuity (section 3.2). However, that method is relatively deprecated, as emphasized in the original Wasserstein GAN paper, “weight clipping is a clearly terrible way to enforce a Lipschitz constraint”. While the authors are aware of the case and are conscious about their different use case, nevertheless, alternative ways such as gradient penalty (Improved Training of Wasserstein GANs by Gulrajani et al 2017, aka., WGAN-GP) may be worth considering.
>
> Response: We agree that gradient penalty might lead to model improvements. Since we didn’t observe that weight clipping adversely affected the training procedure of our model before, we focused on weight clipping for the current results. Because of the time constraints of the revision process, it’s hard to find the best way to apply gradient penalty methods to the Surreal-GAN framework and retest models in all different cases at this stage. However, this is in the plan of our further extension of the Surreal-GAN model.
>
> 3. The baseline that the authors compared against are not very competitive. It might be beneficial if a supervised method (is it possible?) can be included, perhaps to set a “practical upper-bound” so that we can see how far the gap there is between that and the proposed method.
>
> Response: We want to thank the reviewer for this suggestion. As in our explanation to comment 4 from reviewer 1 and comment 3 from reviewer 2, compared methods and other state-of-art unsupervised methods do not optimally utilize information from the (reference) CN data, which is one of the important limitations preventing them from providing ground truth disease-specific pattern representation. Based on the reviewer’s suggestion, we compared our model with supervised regression methods (LR and SVR) through 5-fold cross validation. There is an 0.04-0.07 gap in Pattern c-index between Surreal-GAN results and those most optimal results (section A.11). We considered this gap reasonable, but we still have space to further improve the model. Besides that, we also leveraged the ground-truth information and derived some “practical upper bounds” of compared models for better understanding performances of Surreal-GAN (section A.10). However, we want to emphasize that these upper bounds are derived by letting these methods decompose into more than 3 (M=4-10) components and intensively searching for combinations of 3 which best match the ground truth. In real cases, without ground truth, it will be very hard to know what the optimal number of components is and which components are truly disease-related. In contrast, Surreal-GAN can directly and accurately output disease-specific representations without indices capturing non-disease related variations, which is one of the key advantages of the proposed method.

---

> > ### Author Response · Authors · 2021-11-20
> > **Response to Reviewer QqEE (Part2)**
> >
> > 4. Section 3.3. When the authors mentioned that “the model was trained for at least 100000 epochs and saved until the reconstruction loss is smaller than 0.003 and the monotonicity loss is smaller than 6E-4”, I was surprised by the huge number of epochs. My primary concerns are: 1) does that mean the initial learning rate is a bit too low and/or a bad learning rate scheduling is used? 2) how do you avoid overfitting, besides using the suite of regularizations in the loss function?
> >
> > Response: Thank the reviewer for this important question and the suggestions will definitely help us improve the model. We did more experiments with different sets of initial learning rates. Preliminary experiments did show that higher initial lr lead to faster convergence of the model without affecting representation accuracies. Because of the time limit of the revision process, it is difficult to systematically test different learning rates in all experiments we have done in this paper. Thus, we added an explanation of this issue in section 3.3. Systematic tests of lr are in our plan of extension to the current model and a better selection of lr will be done when applying this model to different datasets. \
> > Regarding the overfitting problem, we want to say that it is actually not a big issue for this semi-supervised method (actually unsupervised with respect to R-indices). In five-fold cross validation results (Table 6), we can see that the model almost achieved the same performances on train and test set. Besides that, in real data experiments, we randomly half split the AD data into train and test sets. The model derived from the training set was applied to derive R-indices for test data. Voxel-wise comparison results (Fig. 5) using test data replicate the results we found with the whole dataset. Finally, if we want to further avoid overfitting problems, we can actually run CV experiments on the real data and evaluate agreements among models trained on different portions of the data.
> >
> > 5. Section 2.3. The full loss formula is indexed twice (14 and 15). Is it a mistake?
> >
> > Response: Thank the reviewer for catching this mistake. We have corrected the indices in the manuscript now.

---

> > > ### Comment · Reviewer_QqEE · 2021-11-29
> > > **Comment to author response**
> > >
> > > Thank you a lot for writing the detailed rebuttal and for further improving the manuscript. I sincerely hope some of my comments are actually helping. Besides, it's beyond my expectation that you conducted the ablation study which I commented as a "plus" rather than a "requirement".
> > >
> > > I will increase my recommendation score from 6 to 8.

---

> > > > ### Author Response · Authors · 2021-11-29
> > > > **Response to Reviewer QqEE**
> > > >
> > > > Your comments are very helpful for us to improve the paper and the model. Thank you very much!

---

### Official Review · Reviewer_jVuo · 2021-11-02

**Correctness:** 3
**Technical Novelty And Significance:** 1
**Empirical Novelty And Significance:** 2
**Recommendation:** 5
**Confidence:** 4

**Main Review:**

Strengths:

(1) The proposed method models disease as a continuous process and learning infinite transformation directions from CN to PT.

(2) Several key components(function continuity, transformation sparsity, and inverse consistency) can guide the model to capture meaningful imaging patterns).

(3) The experimental results have shown the effectiveness of the proposed model.



Weaknesses:

(1) The overall objective function includes several hyper-parameters, and the authors set fixed values for some parameters. Thus, how to set them? The effects of different settings should be discussed. Besides, when applying to other datasets, whether we can use the same parameters?

(2) As a significant extension of Smile-GAN, the authors should compare the proposed model with Smile-GAN.

(3) In Table 1, the compared methods are out of date, thus more state-of-the-art methods should be included.

(4) It is interesting to validate AD diagnosis performance when using the proposed representation learning method.

(5) The authors state that there are four novelties in this study, however, some of them only introduce the existing technologies into the proposed framework.


**Summary Of The Paper:**

This paper proposes a Surreal-GAN method for learning representations of underlying disease-related imaging patterns. This model has overcome limitations in previously published semi-supervised clustering methods and shown great performance on semi-synthetic data sets.

**Summary Of The Review:**

This paper proposes a Surreal-GAN method for learning representations of underlying disease-related imaging patterns. However, the comparison experiments are not sufficient, and the novelty section needs to be improved.

---

> ### Author Response · Authors · 2021-11-20
> **Response to Reviewer jVuo (Part1)**
>
> 1. The overall objective function includes several hyper-parameters, and the authors set fixed values for some parameters. Thus, how to set them? The effects of different settings should be discussed. Besides, when applying to other datasets, whether we can use the same parameters?
>
> Response: We appreciate the reviewer for these important questions. Indeed, model selection and hyper-parameter tuning are non-trivial, especially in deep learning. To address this, in section A.7, we evaluated the model with varying hyper-parameter values. In conclusion, model performance in different tasks were shown to be robust to parameter values selected from 50%-150% of preset values (introduced in section 3.3). Moreover, since agreements among models are shown to be indicative of representation accuracies, pattern-agr-index can be used as a metric for parameter selections if the user wants to further refine hyper-parameters when applying the model to a different dataset.
>
> 2. As a significant extension of Smile-GAN, the authors should compare the proposed model with Smile-GAN.
>
> Response: As in our response to the comment3 from reviewer 1, Smile-GAN and Surreal-GAN are not directly comparable since they seek different end points. In particular, Smile-GAN, with categorical latent variable aims to cluster patients into a hard categorical subtype membership, whereas Surral-GAN dissects the heterogeneity into a continuous representation, with both an continuous z variable and various additional but effective regularization terms. Completely different structure of results from two methods make it impossible to directly compare Surreal-GAN with Smile-GAN. With that being said, we also studied the effect of the additional regularization terms by comparing Surreal-GAN with  a pseudo comparable “Smile-GAN” (Using continuous latent variable with same regularizations as Smile-GAN model). With all additional regularization terms removed, pseudo comparable “Smile-GAN” shows much worse performance than the Surreal-GAN model (Fig. 4c). Again, we want to emphasize that, even without these regularization terms, the model is still significantly different from the original  Smile-GAN model. To make it clear, we refer to it as the “pseudo Smile-GAN framework” above and in the appendix, and we provided more explanation on differences between Surreal-GAN and Smile-GAN in section 2 and 5.1.

---

> > ### Author Response · Authors · 2021-11-20
> > **Response to Reviewer jVuo (Part2)**
> >
> > 3. In Table 1, the compared methods are out of date, thus more state-of-the-art methods should be included:
> >
> > Response: We thank the reviewer for pointing out this important issue. As in our response to comment4 from reviewer 1, both compared methods and other state-of-art unsupervised methods do not optimally utilize information from the (reference) CN data, which is one of the important limitations preventing them from providing ground truth disease-specific pattern representation. This is important because our approach effectively focuses on heterogeneity of differences between the reference CN group and a target group of interest, i.e. focuses on heterogeneity of disease effects, rather than heterogeneity depending on various disease-unrelated factors. Also, because of this reason, state-of-art unsupervised methods actually do not show performances better than basic methods in this specific simulation setting. We included two other more advanced models (Discriminant-NMF and opNMF) which may be more suitable for this problem, but they also showed worse performance in comparisons (Table1). Of note, the family of Discriminant-NMF methods keep discriminant power by minimizing within class variations and maximizing among-class variations, thus being not ideal to capture heterogeneity within the PT class. There are also other state-of-arts semi-supervised methods designed specifically for capturing disease-related heterogeneity as referred in the introduction of the paper. However, they aim to cluster patients into a hard categorical subtype membership, ignoring that disease involved as a continuum. Thus, with different end points, these methods can not be directly compared with Surreal-GAN same as the Smile-GAN model. We have added more discussion on this in section 5.1. To the best of our knowledge, Surreal-GAN is the first semi-supervised representation approach for learning disease-related heterogeneity, and thus it is hard to conduct fairer model comparisons. Thus, we leveraged the ground-truth information and derived some “practical upper bounds” of compared models for better understanding performances of Surreal-GAN (section A.10). However, we want to emphasize that these upper bounds are derived by letting these methods decompose into more than 3 (M=4-10) components and intensively searching for combinations of 3 which best match the ground truth. In real cases, without ground truth, it will be very hard to know how many components are optimal and which components are truly disease-related. In contrast, Surreal-GAN can directly and accurately output disease-specific representations without indices capturing non-disease related variations, which is one of the key advantages of the proposed method.  Also, we compared our methods with supervised regression models (the most optimal model) to understand the space for improvements (section A.11).
> >
> > 4. It is interesting to validate AD diagnosis performance when using the proposed representation learning method.
> >
> > Response: We really appreciate this interesting suggestion. We used the derived 2D R-indices as input features and ran cross validation with the SVM model for PT/CN classification. Classification results were further compared to that derived with 139 ROIs as input features (section A.13). There is only a mild drop in classification AUC values (from 0.929+-0.015 to 0.895+-0.011), which means that 2D R-indices preserve almost all discriminant information in the original 139 ROI volumes, but significantly improves the model’s interpretability in clinical settings.
> >
> > 5. The authors state that there are four novelties in this study, however, some of them only introduce the existing technologies into the proposed framework.
> >
> > Response: We apologize for this problem. Our idea was to emphasize that these are some important components/changes in the model, though they might share some common ideas with previous works. We have changed our language when describing these parts of the model.

---

### Official Review · Reviewer_Rbz3 · 2021-11-02

**Correctness:** 3
**Technical Novelty And Significance:** 4
**Empirical Novelty And Significance:** 2
**Recommendation:** 8
**Confidence:** 4

**Main Review:**

The model proposed provides an interesting idea of using GAN to create an inverse function to detect the pattern and severity of disease from real data. The regularizations suggest many ways it could be used for other types of data that has a continuous output space. The reasoning behind the regularizations is overall reasonable. There could be more explanation, however, on why double decomposition is a good idea for the model.
Regarding the experiment with simulated data, it is hard to understand why the number of patterns were chosen to be 3, unlike 2 which was suggested to be optimal from the real data. Also, since the regularizations are the major extensions of the model from Smile-GAN, it would be more comprehensible for the reader if the paper had cases where some parameters were set to 0 (where some regularizations were not used). If the regularizations indeed created the effect the authors propose them to, removing them one by one would show the significance of each loss.
The paper also compares between other methods, without a reasonable explanation on why they were selected for comparison. Other references on cases where these approaches were used for the same goal would be helpful.
Some things that could be considered would be using W-GAN’s loss. It has been proposed to solve some issues of GAN by making the model architecture more stable and easier to test (Arjovsky, Chintala, Bottou, 2017), which is useful in generalization of the model or training using different datasets. Additionally, there are words that gradient clipping is a better way than weight clipping to ensure Lipschitz continuity as it still can suffer problems from having a wrong weight clipping window.
One trivial thing is Figure 2.b axis description is cut.

**Summary Of The Paper:**

Surreal-GAN aims to create fake pathological data with a latent variable and a healthy input and includes an inverse function that predicts the latent variable from a fake/real pathological data. However, unlike its predecessor model Smile-GAN, this adds 5 regularizations (L1 loss between generated output and input, L2 loss between the latent variable and the reverse generated latent variable, decomposing the latent variable, making the latent variables orthogonal to each other, and adding positive correlation to disease pattern and the latent variable) to make the latent variable more interpretable and continuous. This would increase the interpretability since instead of clustering the real data it could be used as understanding how severe the patient’s disease progress is. The results show that the model has a higher c-index than NMF, LDA and FA model predictions. It also shows that the best number of patterns for the model to predict is two, and it indeed has positive correlation to the severity of the disease.

**Summary Of The Review:**

Overall, the paper is convincing and has reasonable suggestions in creating a correlated continuous latent space for pathological data. However, there could be more comparisons important in proving their case. Also, there could be more considerations on new ways the model could be developed, given how GAN has been evolving over the years.

---

> ### Author Response · Authors · 2021-11-20
> **Response to Reviewer Rbz3 (Part1)**
>
> 1. There could be more explanation, however, on why double decomposition is a good idea for the model.
>
>
> Response: Based on the reviewer's description, we think that the reviewer is referring to the double sampling procedure. The rationale behind is that we sought to control the monotonicity of the modelling. Specifically, it serves as an straightforward and effective way to avoid oscillation in regional volumes as values of z components increase. It might also be possible to boost a positive correlation between values of z and synthesized changes, but it can not strictly control changes in each ROI. To support our hypothesis, from supplementary experiments in section A.8, we can clearly see the contribution from monotonicity loss enabled by the double sampling procedure.
>
> 2. Regarding the experiment with simulated data, it is hard to understand why the number of patterns were chosen to be 3, unlike 2 which was suggested to be optimal from the real data.
>
> Response: We thank the reviewer for raising this point. Surreal-GAN is an unsupervised representation learning without known ground truth, i.e., the number of patterns. We aimed to make the simulation more realistic and complex enough (3>2) so that the model can be robust in real clinical application. We chose the number of patterns to be 3 because 2 patterns might be an easy case for testing the model. We expected this model to be utilized on different datasets in the future with an unknown number of patterns, though it was only applied to the AD data in this paper. Therefore, a harder case with three patterns might be a better choice for synthetic experiments.
>
> 3. Also, since the regularizations are the major extensions of the model from Smile-GAN, it would be more comprehensible for the reader if the paper had cases where some parameters were set to 0 (where some regularizations were not used). If the regularizations indeed created the effect the authors propose them to, removing them one by one would show the significance of each loss.
>
> Response: We really appreciate this constructive suggestion and we agree that this experiment will help readers and us better understand contributions from different regularizations. Based on the suggestion, we conducted ablation experiments on the basic case and the mild change case (section A.8). In both cases, we can quantitatively visualize and quantify contributions from each regularization term. In nature, Smile-GAN and Surreal-GAN are not directly comparable since they seek different end points, where Smile-GAN, with categorical latent variable, aims to cluster patients into a hard categorical subtype membership, whereas Surreal-GAN captures the heterogeneity via a continuous representation, with both an continuous z variable and various additional but effective regularization terms. Completely different structure of results from two methods make it impossible to directly compare Surreal-GAN with Smile-GAN. With that being said, we also studied the effect of the additional regularization terms by comparing Surreal-GAN with a pseudo comparable “Smile-GAN” (Using continuous latent variable with same regularizations as Smile-GAN model). With all additional regularization terms removed, pseudo comparable “Smile-GAN” shows much worse performance than the Surreal-GAN model (Fig. 4c). Again, we want to emphasize that, even without these regularization terms, the model is still significantly different from the original  Smile-GAN model. To make it clear, we refer to it as the “pseudo Smile-GAN model” above and in the appendix.

---

> > ### Author Response · Authors · 2021-11-20
> > **Response to Reviewer Rbz3 (Part2)**
> >
> > 4. The paper also compares between other methods, without a reasonable explanation on why they were selected for comparison. Other references on cases where these approaches were used for the same goal would be helpful.
> >
> > Response: We appreciate the reviewer for pointing out this limitation of the paper. We have elaborated more on why these methods were selected for comparisons in section 4.2. Besides that, we also discussed limitations of these methods for solving this specific problem (section 5.1). In essence, the compared methods and other state-of-art unsupervised methods do not optimally utilize information from the (reference) CN data, thus can not specifically represent disease-related variations we simulated in the synthetic test. This is important because our approach effectively focuses on heterogeneity of differences between the reference CN group and a target group of interest, i.e. focuses on heterogeneity of disease effects, rather than heterogeneity depending on various disease-unrelated factors. There are some other semi-supervised methods (Varol et al. (2016), Dong et al. (2015), Wen et al. (2021), Yang et al. (2021)) which leverage CN-PT domain mapping information and are recently applied for distangling disease related heterogeneity. However, they aim to cluster patients into a hard categorical subtype membership, ignoring disease involved as a continuum. Thus, same as the Smile-GAN model, with different end points, these methods can not be directly compared with Surreal-GAN. We have included references and discussions to these methods in the introduction and added more discussion in section 5.1. Since, to the best of our knowledge, Surreal-GAN is the first method aiming to learn disease-specific representations, we leveraged the ground-truth information and derived some “practical upper bounds” of compared models for better understanding performances of Surreal-GAN (section A.10). However, we want to emphasize that these upper bounds are derived by letting these methods decompose into more than 3 (M=4-10) components and intensively searching for combinations of 3 which best match the ground truth. In real cases, without ground truth, it will be very hard to know how many components are optimal and which components are truly disease-related. In contrast, Surreal-GAN can directly and accurately output disease-specific representations without indices capturing non-disease related variations, which is one of the key advantages of the proposed method.
> >
> > 5. Some things that could be considered would be using W-GAN’s loss. It has been proposed to solve some issues of GAN by making the model architecture more stable and easier to test (Arjovsky, Chintala, Bottou, 2017), which is useful in generalization of the model or training using different datasets.
> >
> > Response: Thank the reviewer for this suggestion. We did previously test W-GAN’s loss for our model. However, it does not lead to better performances of Surreal-GAN in our preliminary experiments, but we plan to make more efforts for future work to systematically study this, due to the limited time during this revision.
> >
> > 6. Additionally, there are words that gradient clipping is a better way than weight clipping to ensure Lipschitz continuity as it still can suffer problems from having a wrong weight clipping window.
> >
> > Response: We are indeed aware of these different methods for ensuring Lipschitz continuity as discussed in section 3.2. However, since we didn’t observe that weight clipping adversely affected the training procedure of our model before, we focused on weight clipping for the current results. Because of the time constraints of the revision process, it’s hard to find the best way to apply gradient penalty to the Surreal-GAN framework and retest models in all different cases at this stage. However, this is in the plan of our further extension of the Surreal-GAN model.
> >
> > 7. One trivial thing is Figure 2.b axis description is cut.
> >
> > Response: Thank the reviewer for catching this mistake. We have revised the figure now.

---

### Decision · Program_Chairs · 2022-01-20

**Decision:**

Accept (Poster)

**Comment:**

The authors present a GAN for learning a continuous representation of disease-related image patterns from regional volume information generated from structural MRI images.
The reviewers find the problem relevant and appreciate the proposed solution. They find the paper well-written and find the empirical results on Alzheimer brain MRIs relevant for the neuroscience community.

The overall objective function includes several hyper-parameters. As pointed out as the main weak point by multiple reviewers this may hint at overengineering/overfitting to a data set. However, the reviewers also mention that the regularizers are all sufficiently well-motivated in the paper and the author response.

Reviewers highlight comparisons on the real data as a strong result demonstrating that Surreal GAN was able to isolate two major sources/locations of atrophy in Alzheimer’s disease. Overall, the reviews are positive in majority.